# Pharmaceuticals Market, Consumption Trends and Disease Incidence Are Not Driving the Pharmaceutical Research on Water and Wastewater

**DOI:** 10.3390/ijerph18052532

**Published:** 2021-03-04

**Authors:** Omar Israel González Peña, Miguel Ángel López Zavala, Héctor Cabral Ruelas

**Affiliations:** Tecnologico de Monterrey, School of Engineering and Science, Av. Eugenio Garza Sada Sur No. 2501, Col. Tecnológico, Monterrey 64849, Mexico; cabral.hector@tec.mx

**Keywords:** drug consumption, drug market, health risk, pharmaceuticals research, public policies on drug disposal in wastewater, pharmaceutical removal, emerging pollutants

## Abstract

Pharmaceuticals enhance our quality of life; consequently, their consumption is growing as a result of the need to treat ageing-related and chronic diseases and changes in the clinical practice. The market revenues also show an historic growth worldwide motivated by the increase on the drug demand. However, this positivism on the market is fogged because the discharge of pharmaceuticals and their metabolites into the environment, including water, also increases due to their inappropriate management, treatment and disposal; now, worldwide, this fact is recognized as an environmental concern and human health risk. Intriguingly, researchers have studied the most effective methods for pharmaceutical removal in wastewater; however, the types of pharmaceuticals investigated in most of these studies do not reflect the most produced and consumed pharmaceuticals on the market. Hence, an attempt was done to analyze the pharmaceutical market, drugs consumption trends and the pharmaceutical research interests worldwide. Notwithstanding, the intensive research work done in different pharmaceutical research fronts such as disposal and fate, environmental impacts and concerns, human health risks, removal, degradation and development of treatment technologies, found that such research is not totally aligned with the market trends and consumption patterns. There are other drivers and interests that promote the pharmaceutical research. Thus, this review is an important contribution to those that are interested not only on the pharmaceutical market and drugs consumption, but also on the links, the drivers and interests that motivate and determine the research work on certain groups of pharmaceuticals on water and wastewater.

## 1. Introduction

Pharmaceuticals are a group of emergent organic compounds that have contributed to enhance our quality of life. The pharmaceutical industry is responsible for the development, production, and marketing of branded and generic pharmaceuticals. In 2014, total pharmaceutical revenues worldwide exceeded 1 trillion United States dollars (USD) for the first time. The market has been growing at an annual rate of 5.8% since 2017. In 2017, worldwide pharmaceutical market revenue was USD 1143 billion and will reach 1462 billion USD in 2021 [1]. The largest fraction of these revenues corresponds to North America due to the leading role of the US pharmaceutical industry. However, during recent years, the Chinese pharmaceutical industry has shown the highest growth rates amongst countries worldwide [1]. Several factors such as reduced taxes and lowered drug prices in the US, a gross domestic product growth greater than 6% in China and India, widespread population ageing and sedentary lifestyles leading to increase chronic disease, industrialized data services in research and development (R&D) enabling the use of clinical trial data in trial simulations, lowered regulatory barriers for new drugs in the US, and high urban pollution levels increasing the incidence of conditions such as asthma are driving healthcare market growth [2].

Musculoskeletal drugs were the largest pharmaceutical market worldwide with 14% of the total in 2017. The second-, third- and fourth-largest markets were those of cardiovascular, oncological, and anti-infective drugs. The fifth-largest market was that of pharmaceuticals for treating metabolic disorders such as diabetes; diseases of the thyroid and pituitary glands will be the fastest-growing segment of the global pharma market by 2021. This segment will grow at 9% per year in the future, following recent growth of 11.6%. The DrugBank database 2019 (version 5.1.3, released on 2 April 2019) contained 13,336 drugs; 10,256 were small-molecule drugs and 1670 were biotech drugs, while 3732 were approved, 2593 were approved small-molecule, 130 were nutraceutical, 6304 were experimental, 205 were illicit and 256 were withdrawn drugs.

In addition to the pharmaceutical market, pharmaceutical consumption worldwide was also growing, partly driven by a growing need for drugs to treat ageing-related and chronic diseases and changes in clinical practice [3]. Consumption of cholesterol-lowering drugs had nearly quadrupled, use of antidepressant drugs doubled and consumption of antihypertensive and antidiabetic drugs nearly doubled in Organization for Economic Cooperation and Development (OECD) countries between 2000 and 2015 [3,4]. Demand for local and imported pharmaceutical products increased as economies grew, and healthcare provision and insurance mechanisms expanded. Not only was demand increasing, but also the diversity of pharmaceutical needs as emerging markets increasingly address noncommunicable diseases (NCDs) already prevalent in stronger economies, including diabetes and hypertension, while communicable diseases that afflict many emerging markets such as acquired immunodeficiency syndrome, malaria and tuberculosis persist [5]. The increase in the global population is also contributing to pharmaceutical consumption.

It is clear that in 2020 the COVID-19 pandemic has modified and will continue modifying the pharmaceutical market and in the coming years in terms of revenues and investment in new chemical and biological entities due to the efforts to generate a greater amount and more effective vaccines again the SARS-CoV-2 virus. Similarly, the consumption trends in the world will present important changes driven by the urgent need of vaccinating world population to decrease the number of infected people and deaths. In this work, COVID-19 pandemic effects on the drugs market and pharmaceutical consumption trends are not discussed because such data are still not totally available.

As the pharmaceutical market grows and consumption increases, the discharge of pharmaceuticals and their metabolites into the environment, including water, also increases due to their inappropriate management, treatment, and disposal. The occurrence and fate of pharmaceuticals in the environment and the water cycle at trace levels (in the range of nanograms per liter to micrograms per liter) has been widely discussed and reported in the literature during the last two decades; pharmaceuticals are now recognized worldwide as an environmental concern and human health risk [6,7,8,9,10,11,12,13,14,15,16,17,18,19,20,21,22,23,24,25,26,27,28,29,30,31,32,33,34,35]. Adverse effects such as endocrine disruption, changes in behavior, chronic toxicity and impacts to nutrient cycling have been caused by pharmaceuticals in aquatic and terrestrial organisms [8,13,17,19,21,36,37,38,39,40]. Synergistic effects of antibiotics and antimicrobials in aquatic ecosystems have also been reported [9,23].

Pharmaceuticals and their metabolites/oxidation products have been detected in wastewater, surface water, sediments, groundwater and drinking water and have numerous routes by which they enter into the water cycle. They can originate from several sources such as domestic and industrial raw wastewater, treated wastewater and sludge from wastewater treatment plants, septic tanks, latrines, hospitals and pharmaceutical manufacturing facilities, aquaculture, livestock farming and landfills [22,25,27,28,29,30,34,41,42,43,44,45,46,47,48,49,50,51,52].

During the last two decades, a great number of studies have been completed with the aim of determining effective and viable methods to remove or degrade pharmaceuticals and their metabolites and oxidation products from wastewater, surface water, groundwater and drinking water. Several reviews can be found in the literature related to the removal of pharmaceuticals in municipal wastewater treatment plants using different treatment methods [20,53,54,55,56,57,58,59,60,61]. Due to the COVID-19 pandemic and the massive vaccination that is occurring and will occur in the coming months worldwide, the number of studies to detect the SARS-CoV-2 virus, the vaccine constituents and pharmaceuticals used to treat the disease will increase importantly. Additionally, different methods to degrade such substances will also be a common issue on research reports. In this work, such issues are not discussed because such studies are currently ongoing.

Advanced oxidation processes have also been applied for degrading pharmaceuticals. Amongst several approaches, one can mention ozonation [62]; chlorination [63]; chemical [64]; reverse osmosis [65]; activated carbon [66]; sonolysis [67]; electrocatalysis [68]; photo-Fenton [69]; photocatalysis [70] and nonthermal plasma [71]. However, only partial degradation of pharmaceuticals (between 14% and 88%), instead of total mineralization, has been reported in previous studies [69,72]. Additionally, in most cases, the reaction times were not sufficiently short [73], the intermediates/oxidation products were not degraded [74,75] and a continuous injection of oxygen to generate H_2_O_2_ was required; thus, these can be considered expensive and ineffective methods.

Recently, technologies such as electrochemical oxidation have shown promising progress due to their versatility, energy efficiency, automation, environmental compatibility, and cost-effectiveness and are currently being used for pharmaceutical removal [76]. However, when electrochemical oxidation is used to degrade pharmaceuticals, potential generation of intermediate/oxidation products has been reported [63,77,78,79,80]. These by-products can have high levels of toxicity and/or can be even more difficult to degrade using typical methods than the initial compound [76]. Fortunately, research worldwide has been addressing this challenge; for instance, it has been successfully oxidized pharmaceuticals and their transformation products using an electrochemical oxidation cell with stainless-steel electrodes [81,82], in which these results are promising for practical applications because short reaction times and low current densities are needed; such current densities can be potentially supplied by photovoltaic cells.

Despite the intensive research work done worldwide in different pharmaceutical research activities such as disposal and fate, environmental impacts and concerns, human health risks, removal, degradation and development of treatment technologies, there has never been an attempt to analyze the relationships among the pharmaceuticals market, the consumption trends, the fatal disease incidence rate and disease burden in the population with respect to the different pharmaceuticals research fronts. Therefore, this paper addresses and discusses the current pharmaceuticals market, the pharmaceuticals consumption trends, the diseases of incidence, and their relationship with the different pharmaceutical research fronts with the aim of finding the drivers and interests that motivate and promote the research on pharmaceuticals.

Additionally, a review of the legislation and regulations available worldwide related to the treatment and disposal of pharmaceuticals into the environment was conducted. The study reveals a growing global pharmaceutical market, a continuous increase of pharmaceutical consumption and an intensive research work on different pharmaceuticals fronts; however, such research is not totally aligned with the market trends and consumption patterns. Furthermore, the legislation and regulations available do not address the treatment and disposal of emergent contaminants, such as the pharmaceuticals; neither these legislations nor regulations promote incentives in favor of the pharmaceutical industry to generate new drugs that can provide adequate treatment for the patient, but also that the drugs have to be friendly to the environment under a philosophy of green chemistry.

As a result, this review study considers in Section 2 the pharmaceutical market, in which the global market, the distribution and revenue are discussed, as well as the profits of the companies with their products of pharmaceutical consumption. Section 3 addresses the causes of death and the burden of disease by relating the gross national income per capita versus the disability-adjusted life years lost due to diseases. In Section 4, information is presented on the studies that have been carried out on the removal of different pharmaceutical groups in wastewater; the activity carried out for these studies is presented by continent and countries most active in research. Section 5 shows the current five research fronts, these are society and public policies (legislation for the removal of drugs in wastewater), wastewater treatment, health risk, detection of drugs in wastewater, and development of environmentally friendly drugs. In Section 6, trends in drug removal studies in wastewater based on methods and technologies are presented. Finally, in Section 7 of conclusions, this review study makes a recapitulation of the topics addressed in order to analyze the coincidences and disagreements of the efforts made by society to achieve sustainability in the treatment of drugs in wastewater. As mentioned above, the information presented and the analysis carried out in this study does not include the high impacts in the short, medium and long term of the COVID-19 pandemic on the pharmaceutical market, consumption trends, disease incidence and research activities in water and wastewater.

## 2. Pharmaceuticals Market

### 2.1. Global Pharmaceutical Market Revenue Distribution

Recently, the pharmaceutical market worldwide has significantly increased. In 2019, the value of the market was on the order of USD 1.25 trillion, in comparison to USD 390 billion in 2001 (Figure 1). The manner by which people obtains and pays for medicine is driven by the pharmaceutical market; however, pharmaceutical companies are well aware that some markets are better than others [83].

North America is the region with the largest portion of pharmaceutical revenue (48.9%), and the US is still leading the pharmaceutical market; however, recently, a group of emerging markets is playing an important role. Emergent economies such as those of Brazil, India, Russia, Colombia, and Egypt are examples of such markets. Despite the participation of Latin American countries in the increasing market, their contribution to global revenue remains insignificant. In contrast, the Chinese pharmaceutical industry has shown the highest growth rates during recent years. Figure 2 shows the projected global pharmaceutical sales for 2022 by region [83]. Much of this prediction lies in the fact that the main pharmaceutical companies that produce drugs are in the regions where the greatest growth is expected as Figure 3 indicates.

### 2.2. Companies and Products

European and American companies remain the leaders of the pharmaceutical market. In 2019, Roche was the company with the largest pharmaceutical revenue on the order of 48.3 billion USD and the top based on research and development (R&D) with 10.3 billion USD; while, in 2018 Pfizer was the world’s largest company with pharmaceutical sales of 45.3 billion USD and 7.96 billion USD on R&D spending and its products were available in more than 125 countries. In 2019, Novartis overcame Pfizer with 46.09 billion USD on sales and 8.39 billion USD on R&D spending. Besides Pfizer (43.66 billion USD on sales and 7.99 billion USD on R&D), other important companies from the US were Merck & Co. (40.90 billion USD on revenue), Bristol-Myers Squibb (40.69 billion USD on sales) and Johnson & Johnson (40.08 billion US dollars of revenue). In Europe, besides Roche and Novartis in Switzerland, Sanofi in France, GlaxoSmithKline and AstraZeneca in the United Kingdom (UK) were the leaders. Surely the report of 2020, will present important changes level of revenues and R&D spending due to the COVID pandemic. Figure 3 shows the top 14 global pharmaceutical companies by sales and R&D spending in 2018 and 2019 [83]. The figure reveals important changes from a year to another and much difference in drug sales among the largest companies in the world; however, the investment they have for R&D does not reflect much discrepancy.

The largest share of pharmaceutical revenue corresponds to branded and patented medicines. Amongst therapeutic drugs, oncologic, antidiabetic, respiratory, autoimmune disease, and antibiotic and vaccine drugs are the top pharmaceuticals generating approximately USD 100 billion, 79 billion, 61 billion, 54 billion and 41 billion, respectively, in 2018 (Figure 4). Individually, the most important pharmaceutical products for revenue generation are Humira (AbbVie, anti-inflammatory), which generated approximately 20 billion USD in 2018; Eliquis (BS/Pfizer, anticoagulant), which produced 9.9 billion USD; and Revlimid (Celgene, immunomodulator agent), which generated 9.7 billion USD. Figure 5 presents the top 15 pharmaceutical products by sales worldwide in 2018 [83]. Most of the companies invest in R&D between 5 and 10 billion USD. However, Humira (AbbVIe) is the one with the highest investment in the U.S. by doubling the amount (20 billion USD) compared to the second place.

Amongst all industries, the pharmaceutical industry has the largest investment on R&D. Such spending involves identification and development of compounds for new drugs, and it is increasing throughout the world over time. Worldwide, the largest number of new compounds and pharmaceuticals between 2013 and 2017 was generated by the US pharmaceutical industry, followed by that of Europe. Figure 6 shows the number of new chemical or biological entities developed between 1992 and 2018, by region of origin [83].

Figure 6 shows that in the period 1998–2002 the Europe and the U.S. developed approximately the same number of compounds in a quantity slightly less than 80. However, the U. S. enhanced the production of new compounds at a higher rate than other regions from 2013. As a result, the U.S. production is around 130 new compounds while Europe remained approximately constant in recent years. This increase has reflected that the U.S. increased its sales compared to other regions. Between 2015 and 2019, the same trend is observed, American companies produced 120 new chemical and biological substances; while Europe and Japan introduced 58 and 36 new products, respectively.

### 2.3. Pharmaceutical Consumption

Similar to the market, worldwide the consumption of pharmaceuticals continues increasing due to changes in clinical practices and the growing demand for drugs for treating ageing-related and chronic diseases. Four groups of pharmaceuticals are the most relevant: cholesterol-lowering, antidepressant, antihypertensive and antidiabetic drugs [3].

In OECD countries, the consumption of cholesterol-lowering drugs nearly quadrupled from 2000 to 2017, as shown in Figure 7. In the OECD countries, an eightfold variation in consumption levels of cholesterol-lowering drugs can be observed; United Kingdom (UK), Denmark, Belgium and Norway had the highest consumption per capita in 2017 [3].

Use of antidepressant drugs doubled in OECD countries between 2000 and 2017 as driven by the recognition of depression, therapeutic treatment, guidelines and changes in patient and provider attitudes [3,4], as shown in Figure 8. However, between countries, a significant variation in consumption of antidepressant can be observed; in 2017, Iceland, Canada, Australia, and the UK had the highest consumption levels, while Latvia, Korea, Hungary and Estonia had the lowest consumption levels [3].

Antihypertensive pharmaceutical consumption nearly doubled in OECD countries from 2000 to 2017, with the highest use occurring in Germany and Hungary at five times the consumption levels of Korea and Turkey (Figure 9). In Luxemburg and Estonia, the consumption levels have nearly quadrupled. These remarkable variations are a consequence of differences in the prevalence of high blood pressure and clinical practices [3].

The use of antidiabetic drugs approximately doubled in OECD countries between 2000 and 2017; Finland, the Czech Republic, Canada, UK, Germany, and Slovenia have the highest consumption, as shown in Figure 10. This increasing consumption can be explained by the increasing prevalence of diabetes and obesity [3].

Regarding the consumption of pharmaceuticals per region, drug consumption varies from region to region; however, some similarities can be observed. The differences and similarities reflect genetic, lifestyle and food diet composition differences and probably differences in food access depending on the population’s economy. From a list of 28 groups of pharmaceuticals, the following analysis was conducted per region. In Europe, the most consumed groups of pharmaceuticals are those treating cardiovascular system, alimentary tract and metabolism, nervous system, agents acting on the renin–angiotensin system and blood and blood-forming organs, whereas antacids are the less commonly consumed. Notably, since 2015, an important reduction in the consumption of all pharmaceuticals has been observed (Figure 11) [3].

In North America (US and Canada), the five most commonly consumed groups of drugs are those for treating the cardiovascular system, agents acting on the renin-angiotensin system, nervous system, alimentary tract and metabolism and lipid-modifying agents; meanwhile, the cardiac glycosides are less commonly consumed (Figure 12). As shown, the four groups of pharmaceuticals are also the most consumed in Europe but in a different order of consumption.

Similar to Europe and North America, in Asia, the most commonly consumed group of pharmaceuticals is that to treat the alimentary tract and metabolism; additionally, drugs for the cardiovascular system, blood and blood-forming organs, nervous system and respiratory system are amongst the most commonly consumed, whereas the cardiac glycoside group is less commonly consumed (Figure 13) [3].

Similar to Europe and North America, in Oceania (Australia and New Zealand), the most commonly consumed groups of drugs are those for treating the cardiovascular system, nervous system and alimentary tract and metabolism; additionally, lipid-modifying agents and agents acting on the renin-angiotensin system are also amongst the most used. In contrast, a less commonly consumed drug group is the antiarrhythmics (Figure 14) [3].

In Latin America, the most consumed pharmaceuticals are agents acting on the renin-angiotensin system, as well as diuretics, drugs used in diabetes, lipid-modifying agents and antidepressants, whereas the antihypertensives are less commonly consumed (Figure 15) [3]. Notably, the analysis for Latin America is not conclusive because limited data are available.

Regarding the consumption of generic pharmaceuticals in OECD countries, in 2017, generics represented more than three-quarters of the total volume consumed in the UK, Chile, Germany and New Zealand, while it represented less than one-quarter in Luxembourg and Switzerland (Figure 16). Some of the differences can be explained by the market structures, number of off-patent medicines and prescribing practices; however, generic consumption also depends on policies implemented by the countries [3,83,84,85].

Some countries have enhanced their efforts to increase generic consumption since the onset of the economic crisis in 2008. Amongst such efforts can be mentioned financial incentives for physicians, pharmacists, and patients.

Notably, the previous analysis only includes pharmaceutical consumption trends in OECD countries because there is limited information regarding the remaining countries; however, such analysis could be considered as representative of global pharmaceutical consumption. As shown, only drugs for treating the cardiovascular system, metabolic disorders and diabetes are on the list of the five most important pharmaceuticals from a market and revenue generation perspective. This is important because the market and revenue generation, and pharmaceutical consumption, might be governed by different drivers. Therefore, it is appropriate to incorporate in this analysis the information on the causes of mortality and burden of the disease shown in Section 3.

## 3. Causes of Death and Burden of Disease

According to the 2017 world census, 57 million people died [86]. Thus, the census shows that the causes of death have a correlation with income levels and that they change from country to country.

Figure 17 shows the number of deaths by cause throughout the world for 2017 [85]. From the information presented in this figure, it is possible to distinguish that cardiovascular disease, anticancer, respiratory system and psychiatry-neurological pharmaceutical drugs are more related to the most typical causes of death. As a result, the world requires production of more of these medications related to the diseases presented in Figure 17.

According to the census presented by H. Ritchie and M. Roser [86], the Global Burden of Disease assessment assigns each death to one specific cause; here, the risk health outcomes and the disease burden are closely related to the variation in risk factors. For example, it can be linked to four broad risk categories: behavioral, environmental, occupational, and metabolic risks. Figure 18 shows the number of deaths by risk factors throughout the world during 2017 [86]. Figure 18 shows that the leading causes of risk are high blood pressure, smoking, high blood sugar, air pollution and obesity. These factors are related to the pharmaceutical drugs addressing cardiovascular diseases, the respiratory system, and metabolic disorders. As a result, this ranking of risk factors associated with deaths does not correspond with the research efforts of the studies published by country. For example, there is information of the publication by country in the Section 4.2. “Research Trends by Country” in this review article, in which one can note that the number of publications related to the removal of anti-infectives and anti-inflammatories pharmaceutical drugs is predominant against that of the other pharmaceutical groups. Thus, Figure 18 shows that, from the perspective of the number of deaths by risk factor, academia and industry have different agendas or priorities according to the number of scientific reports found in Scopus related to the removal of pharmaceuticals from wastewater. Additionally, there are not sufficient statistics regarding the number of deaths by risk factor throughout the world. Moreover, in the world census report by M. Roser and H. Ritchie [86], data are presented by correlating economic factors with diseases. As a result, the data were classified into three groups of diseases as described in Table 1.

From Table 1, Figure 19 shows the total disease burden throughout the world in 2017 as measured using the disability-adjusted life years (DALYs) value in millions. In this context, one DALY equals one lost year of healthy life. In this figure, the diseases in blue are NCDs; the diseases in red are communicable, maternal, neonatal, and nutritional diseases; injuries are shown in grey. Figure 19 almost follows the ranking position of diseases described in Figure 17 (number of deaths by cause). Therefore, the predominant group according to the census is the NCDs; specifically, cardiovascular, cancer and musculoskeletal disorders are the leading groups of the diseases that require more medication throughout the world. Inherently, the Scopus search on musculoskeletal diseases and the removal from wastewater of the pharmaceuticals used to attend such disease did not show results. From the same order of logic, one of the pharmaceutical groups less studied is the anticancer drugs in water treatment; however, according to Figure 19, cancer is one of the most relevant groups of diseases in which more people require medication as well as for cardiovascular system-related diseases.

Figure 20 shows the DALYs lost due to two major groups: communicable and NCDs. It can be observed that communicable diseases are correlated with average income levels.

In Figure 20, it is very revealing to observe the trend in the correlation of the DALYs lost against gross national income for both groups of diseases (noncommunicable and communicable) as reported by the world census [87].

The main observation shown in Figure 20 is that the countries with a robust economy suffer less impact from diseases. In addition, the group of NCDs is less inhibited by the economic factor.

Figure 21 shows that countries with an extremely low health expenditure have the highest DALY rates. Thus, it is clearly perceived that spending on public policies related to citizens’ welfare results in the wellness of the nation; there is a payoff. Furthermore, the data shown in Figure 21 demonstrate that, for instance, countries such as South Korea, Chile, China, Colombia, Peru, Thailand, the Czech Republic, Poland, and Cuba, to name but a few, are investing less than USD 2000 per capita; however, given this, they have the lowest DALY rates. Consequently, they are demonstrably more efficient in their resource management to obtain the same results of those countries that are spending four times or more of this amount such as Switzerland, the US and Norway for their health system. Figure 21 shows the number of DALYs per 100,000 individuals vs. the health expenditure per capita in USD.

## 4. Pharmaceutical Removal in Wastewater under Study

Information of pharmaceuticals under study worldwide was obtained from the Scopus database. Table 2 summarizes all keywords used in the research chain in Scopus; thus, these keywords used associate to the main groups and type of pharmaceuticals under research by the international community. The data presented correspond to results obtained until the end of December 2020. Classification of pharmaceuticals in Table 2 was also complemented based on the pharmaceutical market information above. As a result, if we would like to present an example of this research, it is possible to show the following exercises: if scientific articles associated to the anti-infectives pharmaceutical group is search in Scopus, then the research chain used was (anti-infectives) or (antibiotics) or (antivirals) or (antifungal) and (pharmaceutical)) + (wastewater) or (water) + (removal) or (remotion). Likewise, in the case of scientific articles of respiratory system pharmaceutical group, the search chain in Scopus was (respiratory system) + (wastewater) or (water) + (removal) due to there was not additional keywords as it is also the case of alimentary tract and metabolism pharmaceutical group. However, it was not the case of the example of anti-infectives. Consequently, the nine pharmaceutical groups described in Table 2, correspond to nine research chains in Scopus.

Figure 22 shows the number of publications in the last 22 years regarding the removal of nine pharmaceutical groups in water and wastewater. As seen, the research of pharmaceuticals worldwide is increasing over approximately the last two decades. Specifically, the most intensive research has been done for anti-infective drugs and anti-inflammatory drugs. During these two last decades, research on pharmaceutical groups of psychiatric-neurological, cardiovascular system, metabolic disorders and respiratory system has also shown an important increase. Nonetheless, the less-studied pharmaceutical groups are anticancer, hematological, and alimentary tract drugs. Some trends are interrupted by a discontinuous line; this means that research was not reported in Scopus platform. In addition, Figure 22 shows that the main pharmaceutical group that is being investigated for its removal in water is the anti-infective of which it is the fourth place of the highest sales but that does not appear in the first places of consumption according to the countries of the OECD. As the second most studied pharmaceutical group for its removal in water is the anti-inflammatories, of which it is the first place of the highest sales but also does not appear in the first places of consumption according to the OECD countries. However, the cardiovascular system and metabolic disorder pharmaceutical groups in which they have demonstrated a similar activity on the research studies of water removal by be approximately in third place according with Figure 22 are also the most sale and most consumed according to the OECD countries. In a debatable way, it is shown that the pharmaceutical groups of respiratory system and psychiatric-neurological are also being very active in their research of removal in water, but they do not appear among the most sold groups but among the most consumed in OECD countries in some regions of the world.

### 4.1. Research Trends by Continent

In Table 3, it is shown the research efforts conducted on pharmaceuticals groups in water and wastewater in different regions of the globe. In the second line of Table 3 it is shown the contribution of each continent on the anti-infective drugs research. Moreover, it is shown that Asia has been the leading continent with approximately 67% of the total research on anti-infective drugs, followed by North America with 22% and Europe with 11%. Oceania, Africa, Central and South America are not contributing at all. Nonetheless, according with the pharmaceutical revenue North America and Europe are the countries with more sales. In addition, there is a discrepancy between be Asia the most active in research of removal of anti-infective, but this pharmaceutical group is not into the top five most consumed in the Asia region.

In the third line of Table 3, it is shown the percentage distributions of articles regarding treatment of anti-inflammatories in water per continent. In this group, Europe has been the leading continent with 52% of the total research studies, followed by Asia with 27% and North America with 21%. Oceania and African studies represent 0% of the total studies in this field. The anti-inflammatories pharmaceutical group is not one of the most consumed drugs according with the OECD countries of Asia, North American or Europe, but it is the most sale according with the drug industries.

In the fourth line of Table 3, it is shown the percentage distribution by continent of articles studying the anticancer drugs in wastewater. In this case, Asia was the leading continent with 52% of the total research studies, followed by Europe with 33% and North America with 15%. Conversely, Oceania and Africa represent 0% of the total studies in the field. The high number of publications in Asia regarding this type of drug might be a result of the high ratio of cancer deaths, which is the second to that of Africa. However, the incident rate is higher in North America and Europe. However, the drugs related to cancer are not into the most consumed according with the countries of the OECD even though their sales represent are amongst the highest, with more revenues for the drug industries mainly because they do not have an economical cost to the market.

Likewise, research regarding anticancer drugs has been increasing during the last 10 years mainly as a result of the growth of cancer cases in worldwide populations. According to the World Health Organization, Europe and Asia have the highest number of cancer cases.

In the fifth line of Table 3, it is shown the percentage distributions of articles regarding treatment of water with cardiovascular system drugs per continent. In this respect, Europe has been the leading continent with 46% of the total research studies, followed by North America with 32% and Asia with 22%. Oceania and Africa represent 0% of the total studies in this field. Moreover, according with the drug market Europe and North America have the more sales in the word as well as the medication related with the cardiovascular system is one of the most sale worldwide and is the top more consumed according with the OECD countries of these regions.

In the sixth line of Table 3, it is shown the percentage distribution by continent in terms of articles studying the psychiatric-neurological drugs in wastewater. North America has been the leading continent with the 39% of the total research studies, followed by Europe with 31% and Asia with 30%. In contrast, Oceania and Africa represent 0% of the total studies in this field. Nevertheless, the medication related to the psychiatric–neurological does not represent a pharmaceutical group with the highest revenue for the drug industry, but it is one of the most consumed according with the OECD countries of these regions.

In the seventh line of Table 3, it is shown the percentage distributions of articles regarding metabolic disorder drug treatment in water per continent. In this respect, North America has been the leading continent with 36% of the total research studies, followed by Europe with 30% and Asia with 27%. Alternatively, Oceania represents 7%. Africa produced no studies at all in this field. Likewise, in this pharmaceutical group, there is a conveniently match between the highly research activity in the removal of these drugs in water, as well as this pharmaceutical group represent one with the most revenue for the drug industry and also this group is one of the most consumed according with the OECD countries worldwide.

In line eighth of Table 3, it the percentage distributions of articles regarding hematological drug treatment in water per continent. In this case, North America has been the leading continent with 39% of the total research studies to, followed by Europe with 30% and Asia with 26%. However, Oceania produced few studies in this subject representing 5%. Africa produced no studies at all in this field. The drugs of this group are not of the most research active for the removal of water; however, it is one of the most consumed according with the OECD countries in Europe and Asia; in terms of the revenue, this pharmaceutical group is not enlisted as relevant.

In the nineth line of Table 3, it is shown the percentage distributions of articles regarding respiratory system drug treatment in water per continent. In this case, North America has been the leading continent with 465% of the total research studies, followed by Europe with 28% and Asia with 27%. Oceania and Africa had very few studies and represented 0%. Moreover, the pharmaceutical group of the respiratory system is not among the list of drugs with the highest profit for the industry nether for the consumption of the OECD countries over the world. Paradoxically, the diseases related to the respiratory system is listed as one of the main in the world mortality.

In the tenth line of Table 3, it is shown the percentage distribution articles from the pharmaceutical group of the alimentary tract and metabolism. In this group, Europe leads the research with 50%, follow with Asia with 33% and Oceania with 17%. North America and Africa have not shown studies in this respect. Later in Table 4, it will be shown that particularly this pharmaceutical group is the least studied.

### 4.2. Research Trends by Country

In Table 4, it is shown the number of journal articles completed by the leading countries throughout the world regarding wastewater treatment of each pharmaceutical group. In this section, the numbers of research studies on pharmaceutical removal in wastewater will presented; however, an explanation regarding how to interpret the tendencies of these numbers or the ranking positions of the countries is more appropriate to describe by taking in consideration the information presented previously in Section 3, such as it is the causes of death and burdens of disease by country.

In the second column of Table 4, it is shown the number of scientific articles that has been completed by the most active countries in the study field of wastewater treatment of anti-infective drugs. In this case, China is leading with more than 1700 papers published, followed by the U.S., Spain, and Iran. Interestingly, highly populated countries such as Brazil and India are also amongst the top active in documenting their studies.

In the third column of Table 4, it is shown the number of scientific articles that has been published by the most active countries in the study field of wastewater treatment of anti-inflammatory drugs throughout the world. In this respect, Spain is leading with more than 120 papers, followed by China, the U.S. and Brazil.

In the fourth column of Table 4, it is shown the number of scientific articles that has been published by the most active countries in the study field of wastewater treatment of anticancer drugs throughout the world. In this respect, China is leading with 17 papers, followed by the U.S. and India in the second place, so Spain is in fourth place.

In the fifth column of Table 4, it is shown the number of scientific articles that has been published by the most active countries in the study field of wastewater treatment of cardiovascular drugs throughout the world. In this respect, the U.S. is leading with more than 52 papers, followed by Spain, China, and Italy.

In the sixth column of Table 4, it is shown the number of scientific articles that has been published by the most active countries in the study field of wastewater treatment of psychiatric-neurological drugs throughout the world. In this respect, the U.S. is leading with more than 70 papers, followed by China, Spain, and Germany.

In the seventh column of Table 4, it is shown the number of scientific articles that has been published by the most active countries in the study field of wastewater treatment of metabolic disorders drugs throughout the world. In this respect, the U.S. is leading with 100 papers, followed by China, the U.K. and Canada.

In the eighth column of Table 4, it is shown the number of scientific articles that has been published by the most active countries in the study field of wastewater treatment of hematological drugs throughout the world. In this respect, the U.S. is leading with 17 papers, followed by Spain, India, and Japan.

In the nineth column of Table 4, it is shown the number of scientific articles that has been published by the most active countries in the study field of wastewater treatment of respiratory system drugs throughout the world. In this respect, the U.S. is leading with more than 80 papers, followed by China, Germany, and the UK with India.

In the tenth column of Table 4, it is shown the number of scientific articles that has been published in the study field of wastewater treatment of alimentary tract and metabolism drugs throughout the world. In this respect, India, Germany, U. K., and Grace are the only countries that have been publish one article each country; as a result, the alimentary tract and metabolism is the less studied pharmaceutical group. In detail, Scopus did show very few results for alimentary tract drugs. This means that this pharmaceutical group is not used as keywords by authors in their articles regarding the removal of pharmaceuticals in wastewater. This occurs even though medication related to the alimentary tract is highly present in studies completed by businesspeople and economist, which show intense activity in the production and consumption of these types of medication throughout the world.

Furthermore, according to the values presented for these pharmaceutical groups in Table 4, one can see that the three most active countries in conducting research on pharmaceutical drug removal in wastewater are China, the U.S. and Spain, followed by India and Germany. Likewise, most of the remaining countries presented in the lists for each pharmaceutical group show roughly the same numbers of studies, such as Iran, Brazil, the U.K., Italy, and Canada. Thus, there is already a correlation between the aforementioned cited countries and those presented in terms of the pharmaceutical market. For instance, from Figure 2, it is possible to distinguish that the projected global pharmaceutical sales show that North America and Europe lead in terms of global revenue for the drug industry; thus, these continents are the most active in terms of research on the removal of pharmaceuticals in wastewater.

Moreover, Table 4 described that the most recurrent studies made for wastewater removal are the pharmaceutical groups of anti-infectives and anti-inflammatories.

## 5. Current Pharmaceutical Research Fronts

In addition to describing the pharmaceutical market, the consumption of drugs worldwide, the causes of death and burden of disease, we mentioned that it is important to describe the research efforts done by the world in the removal of the pharmaceutical groups in water or wastewater. Therefore, in this subject it is relevant to separate the different topics in the research activity of the removal of drugs groups; as a result, the methodology used is shown in Table 5, in which there are five different research fronts to be involved; this engagement research corresponds to the environmental problem of removal of these emerging pharmaceutical pollutants. In this way, for instance, if one wants to find scientific articles in Scopus related with the research from the society and public policies the research chain used was (Public policies) or (legislation) + pharmaceuticals + (water) or (wastewater) or (disposal), as the second column in Table 5 remarks. In the similar way, in Table 5 it is shown the keywords used in the research to find the five research fronts.

From the methodology described in Table 5, it is shown in Figure 23 the research fronts. The results obtained from the Scopus in regards of the research front it is possible to observe that most of the research activity correspond to the pharmaceutical detection, followed by environmental treatment. The R&D of environmentally friendly drugs, the drugs health risk on water and society and public policies (legislation for drug removal in water) are the fronts with less attention. The previous result is expected, because when a drug is created the laboratories first are focus on the fact that the new drug can be effective for the cure of a specific disease to be treated in a patient. Therefore, degradability in an environmentally friendly manner is not a priority. As a result, the efficacy of the drug in the patient dominates the world agenda, leaving in a second term the priority for their health risk due to exposure of the drug in water or to enact laws that prohibit its presence in wastewater. In other words, the free market economic system provides rules so that the mass production of drugs and their sale in the market is favorable without having any restriction on their environmental impact. Likewise, part of the enormous problem in the growth of drug consumption lies in the fact that many countries in the world citizens can buy drugs for their consumption without the need to go to a doctor for their prescription. Equally, in some countries with greater regulation for restricting patient prescriptions, a black market is observed that allows people to access to drugs. Specifically, it is relevant to also take a look of the waste public policy treaties or agreements of water waste regulations as it is shown in Table 6 below.

Table 6 describes various waste regulations and their pharmaceutical approaches. Here one can observe that the laws/regulations are generally focused in wastes generated through production processes instead of the wastes that are produced through drug consumption. As a result, by not incorporating into public policy treaties, protocols and conventions the need to decrease the presence of drugs in water as a society, there is therefore no work agenda for researchers, the economic sector and the government to begin to remove the most consumed medications. Likewise, the lack of this demand for the removal of medicines in water means that the international organizations and the government do not pressure the companies in the pharmaceutical industry to generate environmentally friendly medicines.

Regarding the health risk of drugs specifically in drinking water, studies in the U.K., Australia and the U.S. indicate that appreciable adverse effects are highly unlikely. This due to the concentrations of drugs in drinking water being generally more than 1000 times below the Minimum Therapeutic Dose (MTD), which represents the lowest clinically active dose [102]. However, this is not a reason to rejoice since as mentioned in different market studies, per capita consumption of drugs has been increasing globally and not enough attention is paid to the long-term effects of drug residues on both the population and the environment. In fact, it is estimated that the active ingredients are secreted unchanged through urine and feces after consumption of 30% to 90% of the time, directly entering the environment through wastewater [89]. During water treatment, in most cases, contaminants can be partially removed nonetheless the remaining traces persist in the water effluents [103]. As a result, there is indeed a biological accumulation of these emergent pollutants in the ecosystem.

Although the drugs turn out to be beneficial, this does not exempt them from the potential damage they represent to the flora and fauna through the alteration of ecosystems and the possible long-term adverse effect on human health. Therefore, even more attention must be paid to the production of environmentally friendly drugs and to achieve this it is essential to know the 12 principles of green chemistry and apply them in each developmental process described below by Jordan et al. [104].

(a)Prevention. Before to generate a treating process in waste, it is much better to prevent the generation of it.(b)Atom economy. To maximize the incorporation of the regents used on a final product it is convenient to use methods for synthetic the process to avoid not wanted intermediates or produce final products with unnecessary steps.(c)Chemical syntheses with less dangerous precursors and products. Where it is possible, it is convenient to use less toxic and reactive methods that affect the environment or people.(d)Chemical-safe design. For proper use of chemicals, they must be designed with the aim of being less toxic to the environment and people.(e)Safe auxiliaries and solvents. Solvents used as auxiliaries, or drying agents among others, should as far as possible be used in a reduced form.(f)Search for design in energy efficiency. The use of energy required for chemical processes should be kept to a minimum. The above is possible referring to concepts of thermodynamics where the variables of temperature, pressure, volumes, among others, play important roles for the energy required in the process.(g)Use of renewable inputs. As far as possible, raw materials can be used for the purposes of renewable sources.(h)Derivatives reduction. The use of protective groups such as structural modifications to reduce the production of waste and energy consumption should be avoided.(i)Catalysis. As far as possible, the use of catalysts can reduce waste and the appropriate energy requirements such as selectivity among others.(j)Design for degradation. Synthetic molecules must be designed for adequate decomposition in the environment once used. The foregoing anticipates accumulation as a chronic effect.(k)Real time analysis in preventing contamination. In real time, the supervision of the chemical process must be monitored. The above prevents the production of potential hazardous materials due to a control.(l)Intrinsically safer chemistry in preventing accidents. The reagents, and the stages of the whole process must be safer to reduce accidents and/or reduce the exposure of chemical substances in people and the environment.

After considering the 12 principles, it is evident that from 2001 to 2020 the general revenue in pharmaceutical industries has tripled due to excessive consumption of drugs in people. We observe such examples in the years 2000 to 2015 with the quadruple consumption of drugs associated with cholesterol by people, or double the consumption of drugs to treat depression, hypertension, or diabetes, among others. This upward consumption trend is present in all OECD countries in the world, which is why there is an urgent need to incorporate mechanisms to regulate the production of drugs under a green chemistry guideline in laws and treaties worldwide. Likewise, it is important not to omit that the current consumption of medicines with synthetic processes that are not friendly to the environment can cause side effects in people, causing a chronic affectation that triggers in other diseases or even can cause and increment in deaths or quality of life. As a result, it is necessary to evaluate the green drug production strategy and its impact on the environment [104]. This can be determined with the help of the following factors:The selected synthetic route and its effect on the environment.Consider studies on life cycle assessment and the environmental impact of the drug during the development process. It is essential to select compounds that have the least possible adverse effect on the environment.Jointly assess the possible persistence of the drug in the environment and its toxicity and biodegradability.

While there is no full certainty about the adverse effect of drug residues in the environment, it is proposed in these analyzed exercises to push the social commitment to dedicate an effort devoted to a precautionary approach in which the objective is embrace and increase the acceptance of green chemistry as an essential parameter in the production of future drugs.

## 6. Methods and Technologies for the Removal of Pharmaceuticals

Figure 24, Figure 25, Figure 26, Figure 27, Figure 28, Figure 29 and Figure 30 show the percentage distributions of the research efforts in producing a scientific study according to different wastewater treatments for each pharmaceutical group. Therefore, the information presented in this section corresponds to the collected data of the Scopus database. Consequently, the information method consisted of finding a research chain by using the following keywords in the Boolean operators: wastewater OR water; remotion OR removal and finally, any of the terms of the pharmaceutical groups described in Table 2. As a result, Table 7 it is shown the keywords associated with the removal methods. Moreover, in the Scopus research, we used as filters results only for scientific articles for the period from 1998 to 202 in English.

In Figure 24, the most explored methods to remove anti-infective drugs are the biotreatment processes or flocculation with 29% and activated carbon with 30%. This tendency can be expected due to this method is the most economical to treat general pollutants in wastewater but certainly it is not the most effective. Likewise, the activated carbon has been commonly used in pesticides and other volatile organic pollutants, as well as to eliminating the chlorine added in the first stage treatment, through a reducing chemical reaction. However, this method not always is effective to the removal of drugs in wastewater. the oxidation process has been more effective in removing drugs due to be a method that involves the formation of free hydroxyl radicals at very large overpotentials, which is why the radicals present by drugs are oxidized. In fact, to be more effective the process is common to see the coupling of the oxidation process with ozone processes, ultraviolet radiation, etc. However, it is a method that involves a lot of energy and high costs compared to the other methods.

In Figure 25, the methods more commonly used to remove anti-inflammatory drugs are the biotreatment processes or flocculation with 23%, activated carbon and oxidation processes each with 24% and photo methods with 24%. Here the costliest methods such as oxidation process and photo methods have been shown to be effective with respect to others. However, if other methods are present in scientific studies can be notable due to are less expensive to remove the drug pollutants. In addition, activated carbon has been shown to be widely used for being a well know treatment process for organic compounds.

In Figure 26, the most explored methods to remove anticancer drugs is activated carbon with 28%, followed by oxidation processes with 21%, and photo methods with 21%. Similar to Figure 25, the costliest methods of oxidation process and photo methods are become useful for removal medication related with cancer.

In Figure 27, the methods more commonly used to remove cardiovascular system drugs are the oxidation processes with 19%, biotreatment processes or flocculation with 23%, activated carbon with 20% and photo methods with 23%. Likewise, Figure 27 shows roughly a similar distribution in most of the methods, for which the investigation to remove drugs related to the cardiovascular system is presented as a subject not very widely explored to outline a main methodology as the most effective.

In Figure 28, the methods more commonly used to remove psychiatric-neurological drugs are the biotreatment processes or flocculation with 25%, oxidation processes with 19%, activated carbon with 21%, and photo methods with 22%. Similar to the effect found in Figure 27, Figure 28 also reveals an approximate distribution of the various drug removal methods. Therefore, it is possible that the research to eliminate drugs related to the nervous system has not been widely studied; Thus, a predominant methodology has not been presented as the most effective.

In Figure 29, the methods more commonly used to remove metabolic disorder drugs are the biotreatment processes or flocculation with 28%, membrane processes with 12%, oxidation processes 15%, activated carbon each with 23% and photo methods with 18%. Thus, the percentage distribution of different methodologies is well distributed, which it can be an indicative that the removal of this pharmaceutical group has been explored though all methods almost in the same frequency.

In Figure 30, the methods more commonly used to remove hematological drugs are nonthermal plasma with 29%, membrane processes with 21%, and activated carbon with 19%. For the association of metal ions such as iron in blood as part of the subject to deal with hematological drugs it is possible to see a dominant treatment method such as it is in this case the non-thermal plasma process in Figure 30.

In Figure 31, the methods more commonly used to remove respiratory system drugs are activated carbon with 35% and biotreatment processes or flocculation with 29%. Once more time, the activated carbon has been shown to be widely used to remove drugs related to metabolic disorder for being a well know treatment process for organic compounds. However, an economical method such as it is the biotreatment process or flocculation has been shown to be effective and popular.

In the case of the removal of the pharmaceutical group of alimentary tract and metabolism, there are just three studies in which the removal method used was activated carbon, as it is shown in Figure 32.

## 7. Conclusions

In this work, an attempt was done to analyze the pharmaceutical market, drugs consumption trends and the pharmaceutical research interests worldwide. Notably, this work was a big challenge, first because classification and conventions for naming various pharmaceutical groups are not the same throughout communities; for business people and economists, it is not the same as that for the communities of health professionals or scientists who address chemistry or the environment.

Intensive research work has been conducted worldwide in different pharmaceutical research fronts such as pharmaceutical detection in water and wastewater, disposal, fate, environmental impacts and concerns and environmental treatment, human health risks, and degradation and development of treatment technologies for green chemistry in the production of drugs; however, such research is not totally aligned with the market trends and consumption patterns. There are other drivers and interests that encourage and promote the pharmaceutical removal research in wastewater and water. Thus, this paper is an important contribution to those that are interested not only on the pharmaceutical market, drugs consumption, or disease burden rate; consequently, it is also critical to associate the above-mentioned subjects to the different pharmaceutical research fronts for the drug removal in water and wastewater currently under attention and concern of the scientific community.

It was evidenced in this work that public policy treaties, protocols and agreements have not incorporated a demand or need to decrease the presence of medicines in water, therefore, there is no work agenda for researchers, the economic sector and the government to begin to eliminate most of the most consumed medications. Likewise, the lack of this demand for the elimination of medicines in water means that international organizations and the government do not pressure at least legally to companies in the pharmaceutical industry to generate environmentally friendly medicines. In addition, the ecosystem will soon accumulate a saturation threshold of the concentration of the active components of the drugs due to it is estimated that the active ingredients are secreted unchanged through urine and feces after consumption in 30% to 90% of the cases, directly entering the environment through wastewater. During water treatment, in most cases, contaminants can be partially removed nonetheless the remaining traces persist in the water effluents. Thus, the work agenda must consider conducting research on the health risk link to the drug exposure present in water.

In terms of the pharmaceutical market, the study shown that the most active continents are North America and Europe. This relation is also the same in terms of the research efforts to study the removal of pharmaceuticals in wastewater. However, the study of the treatment of drugs in water has recently become very active in Asian countries such as China that is now leading as a country in conducting such studies. Over the last two decades, the production of scientific articles addressing the study of drug removal in water of the pharmaceutical group of anti-infective drugs has been increasing. Specifically, it has been observed that the most studied pharmaceutical groups for the removal in wastewater and water are anti-infectives, and anti-inflammatories; in contrast, the alimentary tract and metabolism is the less studied pharmaceutical group. In the case of the alimentary tract and metabolism pharmaceutical group, there is a possibility that the research studies are not using ‘alimentary tract’ and/or ‘metabolism’ as a keyword to classify these studies, or this pharmaceutical group can be highly omitted since medication related to the alimentary tract and metabolism is highly present in businesspeople and economist studies, which show intense activity in the production, sales and consumption of world reports.

On the other hand, according to the world census, the priority for removal of drugs should be focused on the study of the removal of cardiovascular system-related drugs because they address the sector with the highest number of deaths. In addition, cardiovascular diseases are the main risk factor for health worldwide. Research fronts are increasingly more focused on the detection of drugs as an emerging pollutant in water, followed by environmental treatments. The more studied water treatment methods, according to this research are biotreatment processes or flocculation, followed by activated carbon, oxidation processes and photo methods.

As a general overview, this work presents that the indicators in the pharmaceutical industry are revenues from drug sales and this has caused an exorbitant consumption by people. As a result, there is a rise in the consumption and sales where there are no effective restrictions to avoid self-prescription, or medical doctors themselves are not having a concern that their patients should consume fewer drugs to treat the same diseases. Therefore, the economic system has governed the priority in profits and not the concern about developing policies to have an economy respecting the sustainability of producing green chemistry drugs, or reducing the risk of exposure of drugs in water by studying more the toxicity of them or by adding more regulations on wastewater discharges by the governments.

Finally, this review study, presented as its main objective, the worrying need to make known to the different sectors of society, government, economic sector and health systems the unpostponable need to act to generate more effective public policies to mitigate the contamination of water by drugs, where currently there is no coincidence, or in any case only partial, between the drugs most studied in their removal by researchers and the most consumed and/or sold drugs. It is important to note that the information presented together with its analysis in this study does not include the high impacts in the short, medium and long term of the COVID-19 pandemic on the pharmaceutical market, consumption trends, disease incidence and research activities in water and wastewater. Nonetheless, when the information would be available it will be interesting to conduct a research to evaluate and address such effects.

## Figures and Tables

**Figure 1 ijerph-18-02532-f001:**
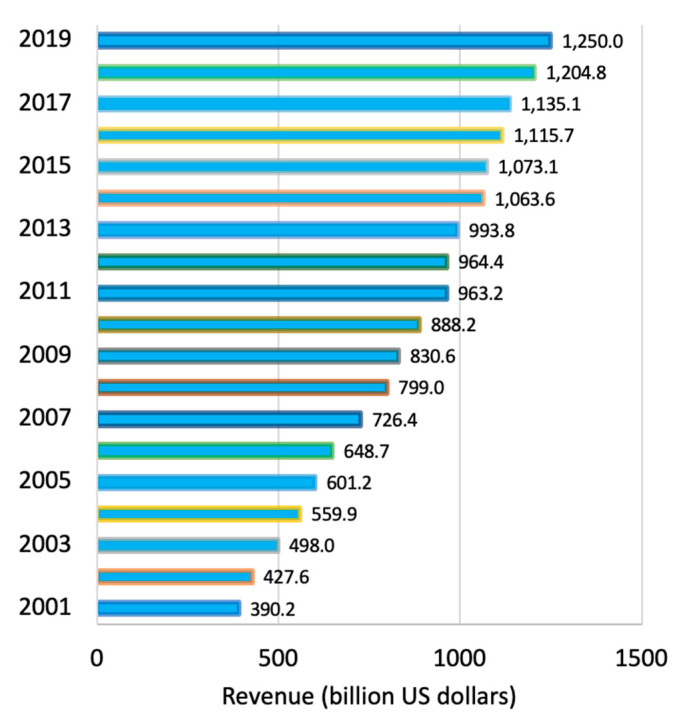
Global pharmaceutical revenue from 2001 to 2019 [83].

**Figure 2 ijerph-18-02532-f002:**
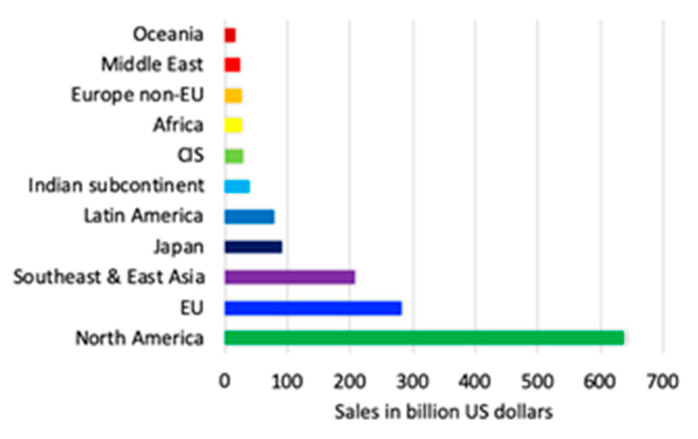
Projected global pharmaceutical sales for 2022 by region [83].

**Figure 3 ijerph-18-02532-f003:**
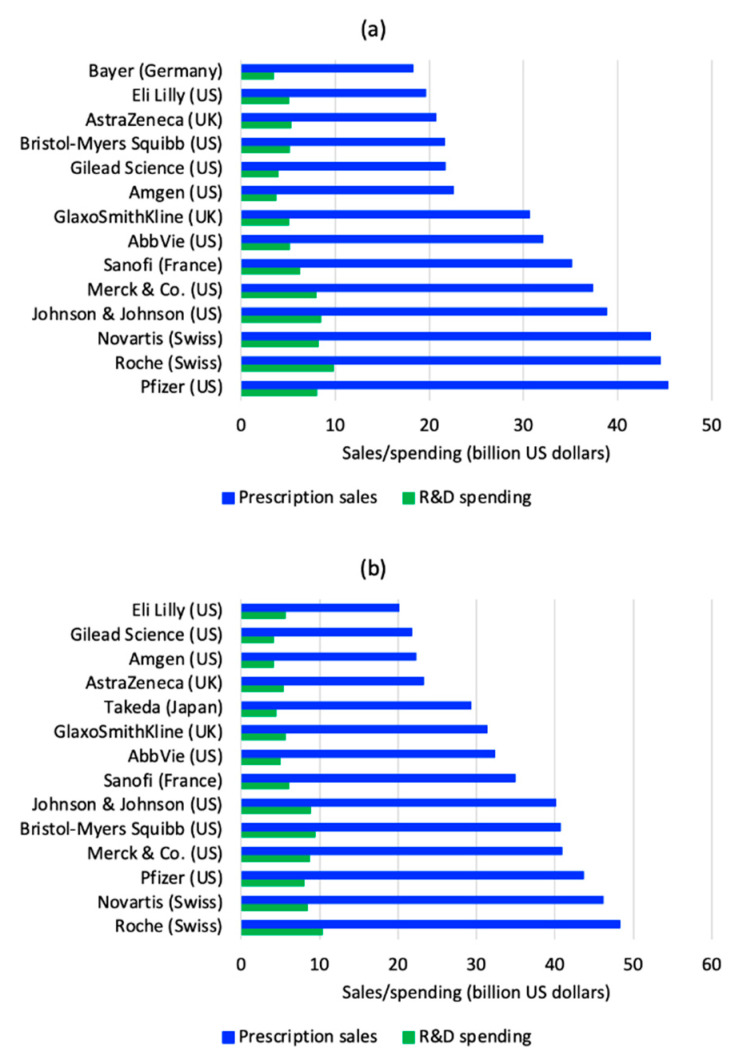
Top global pharmaceutical companies by sales and R&D spending in 2018 (**a**); and 2019 (**b**) [83].

**Figure 4 ijerph-18-02532-f004:**
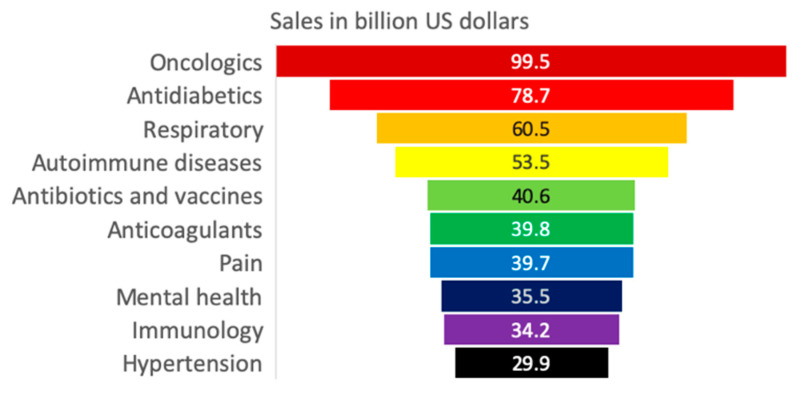
Global revenue by pharmaceutical group [83].

**Figure 5 ijerph-18-02532-f005:**
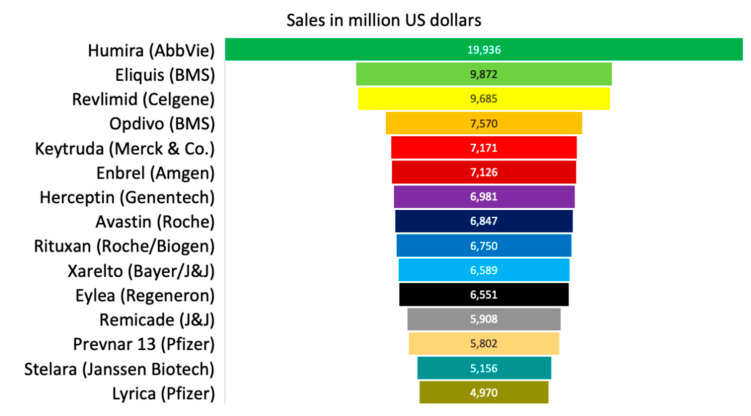
Top pharmaceutical products by sales worldwide in 2018 [83].

**Figure 6 ijerph-18-02532-f006:**
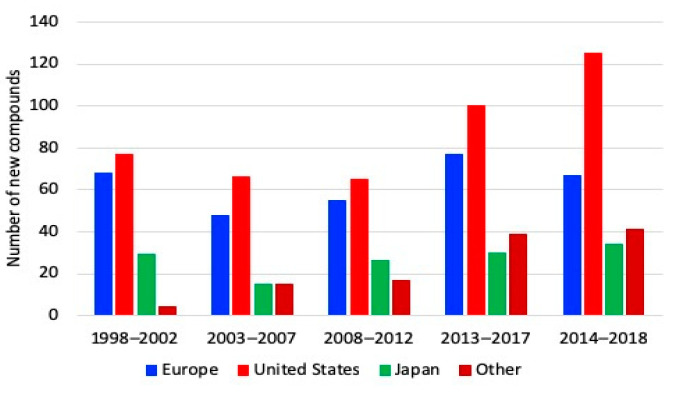
New chemical and biological entities developed between 1992 and 2018 [83].

**Figure 7 ijerph-18-02532-f007:**
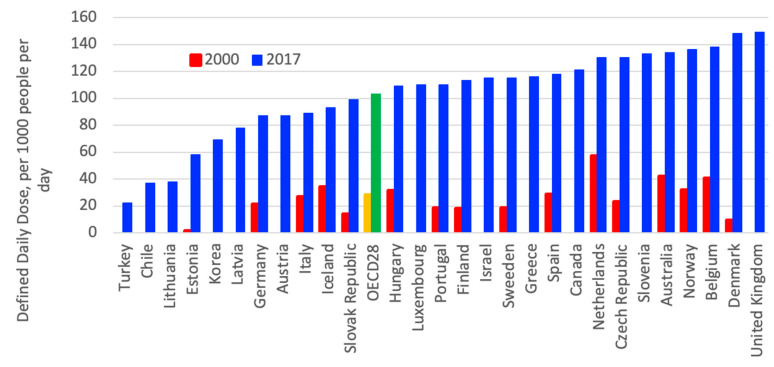
Consumption of cholesterol-lowering drugs in Organization for Economic Cooperation and Development (OECD) countries from 2000 to 2017 [3].

**Figure 8 ijerph-18-02532-f008:**
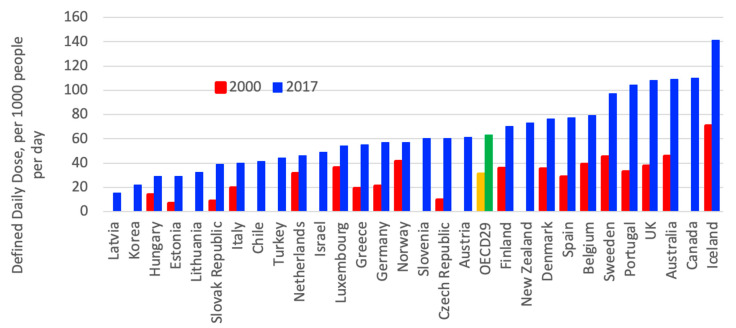
Consumption of antidepressant drugs in OECD countries between 2000 and 2015 [3].

**Figure 9 ijerph-18-02532-f009:**
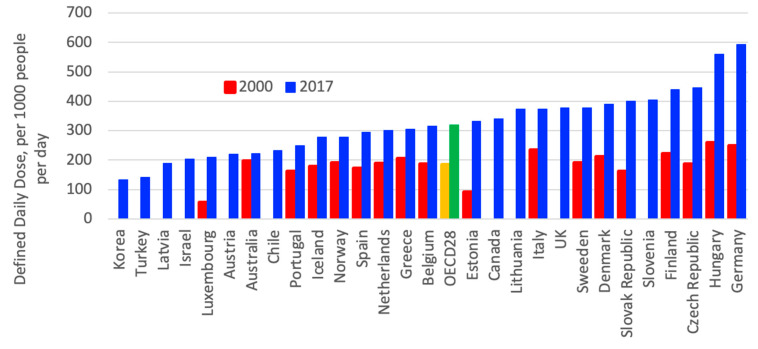
Consumption of antihypertensive pharmaceuticals in OECD countries from 2000 to 2017 [3].

**Figure 10 ijerph-18-02532-f010:**
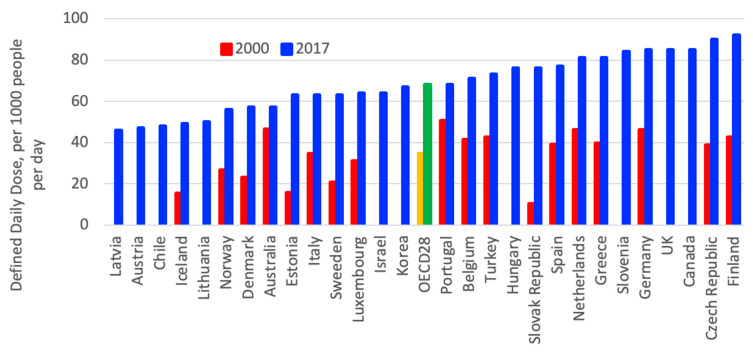
Consumption of antidiabetic drugs in OECD countries from 2000 to 2017 [3].

**Figure 11 ijerph-18-02532-f011:**
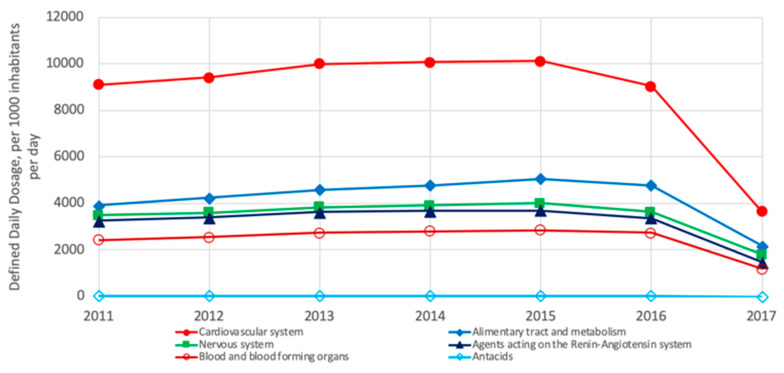
Most consumed pharmaceuticals in OECD European countries from 2011 to 2017 [3]. Drugs for treating the cardiovascular system are most consumed in both regions [3].

**Figure 12 ijerph-18-02532-f012:**
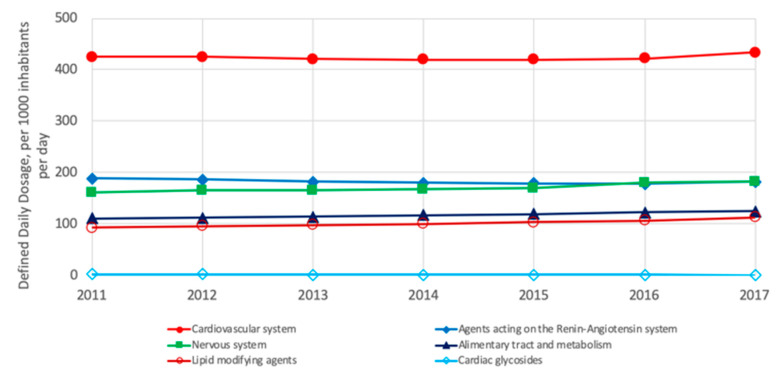
Most consumed pharmaceuticals in the US and Canada between 2011 and 2017 [3].

**Figure 13 ijerph-18-02532-f013:**
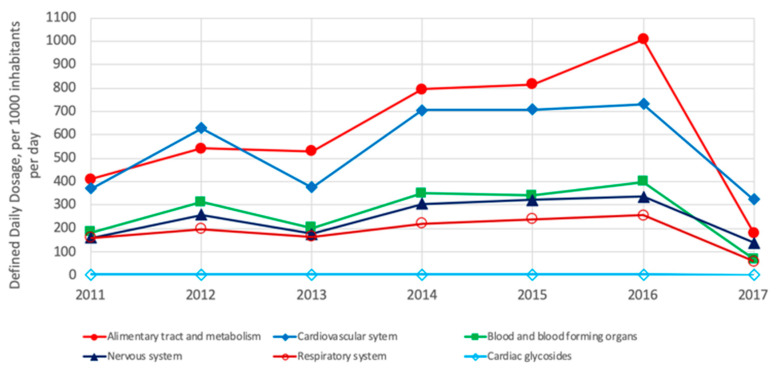
Most consumed pharmaceuticals in OECD Asian countries from 2011 to 2017 [3].

**Figure 14 ijerph-18-02532-f014:**
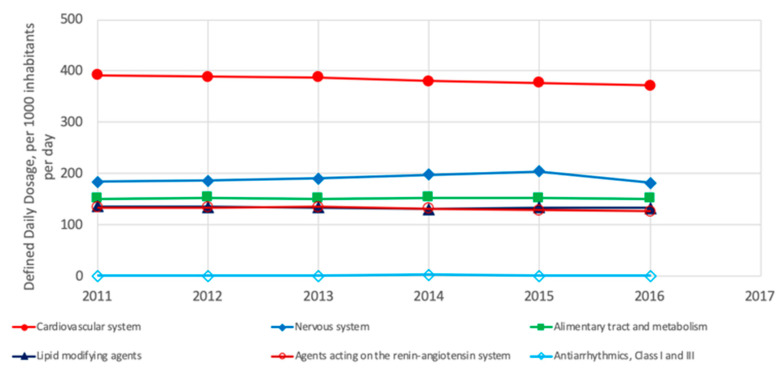
Most consumed pharmaceuticals in Australia and New Zeeland between 2011 and 2017 [3].

**Figure 15 ijerph-18-02532-f015:**
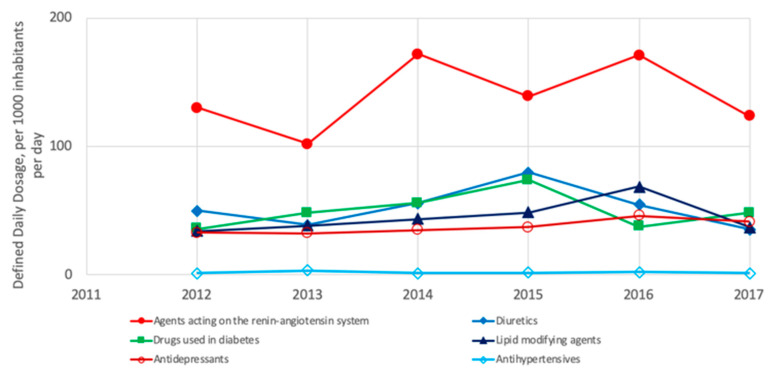
Most consumed pharmaceuticals in OECD Latin American countries from 2011 to 2017 [3].

**Figure 16 ijerph-18-02532-f016:**
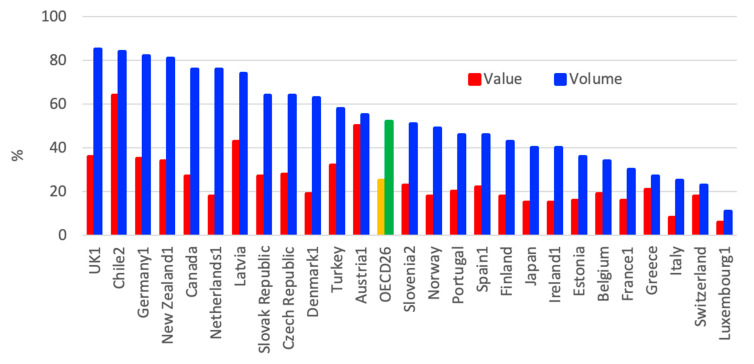
Consumption of generic pharmaceuticals in the OECD countries in 2017. (1) Reimbursed pharmaceutical market. (2) Community pharmacy market [3].

**Figure 17 ijerph-18-02532-f017:**
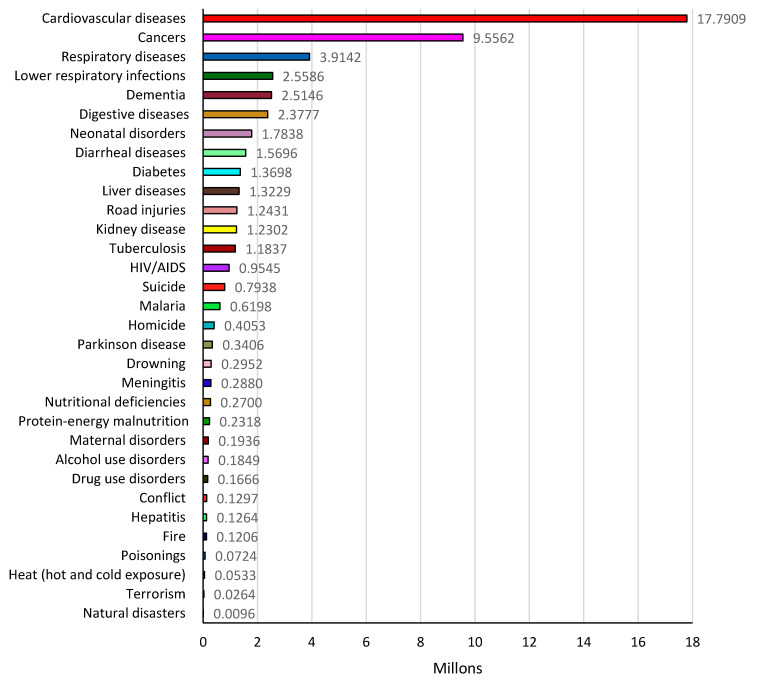
Breakdown of global deaths by cause, world, 2017. The information is provided as the share of annual deaths rather than the absolute number. Information provided by Hannah Ritchie and Max Roser (2019) in “Causes of Death,” published online at OurWorldInData [86].

**Figure 18 ijerph-18-02532-f018:**
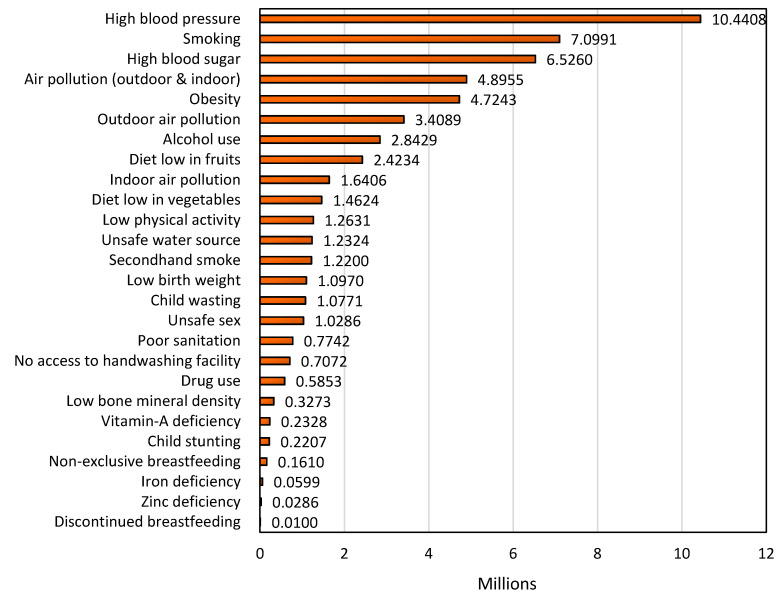
Total annual number of deaths by risk factor, World, 2017. The data were collected measured across all age groups and both sexes. Information provided by Hannah Ritchie and Max Roser (2019) in “Causes of Death,” published online at OurWorldInData [86].

**Figure 19 ijerph-18-02532-f019:**
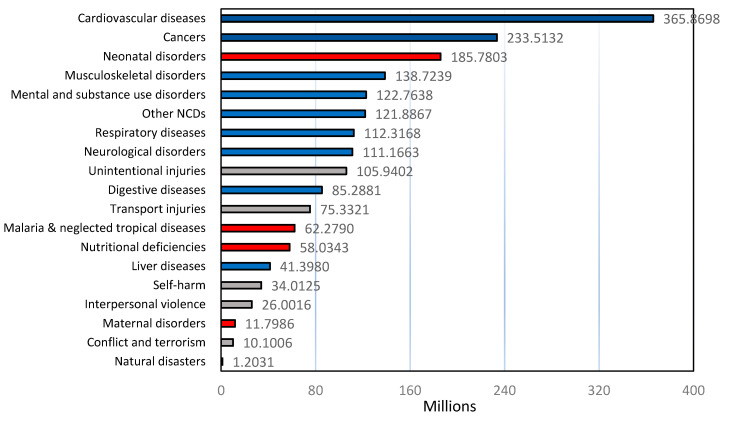
Total disease burden in the world in 2017, measured using the value disability-adjusted life years (DALYs) on millions. One DALY equals to one lost year of healthy life. Information provided by M. Roser and H. Ritchie, “Causes of Death”. Published online at OurWorldInData [87].

**Figure 20 ijerph-18-02532-f020:**
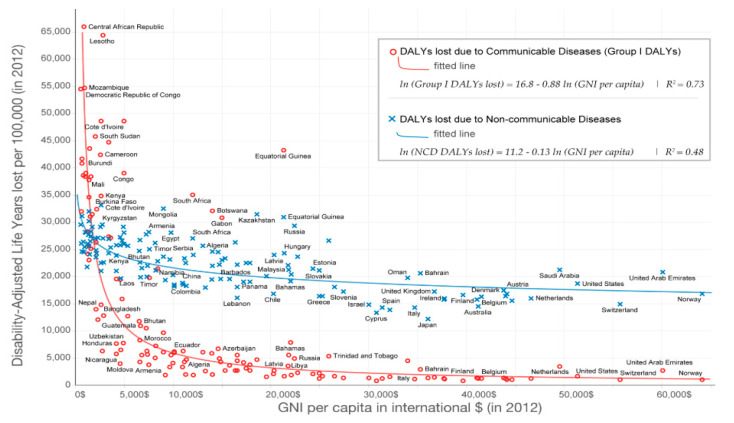
Relationship between gross national income (GNI) per capita vs. DALYs per 100,000 individuals throughout the world in 2012. Information provided by M. Roser and H. Ritchie (2019) in “Causes of Death,” published online at OurWorldInData [87].

**Figure 21 ijerph-18-02532-f021:**
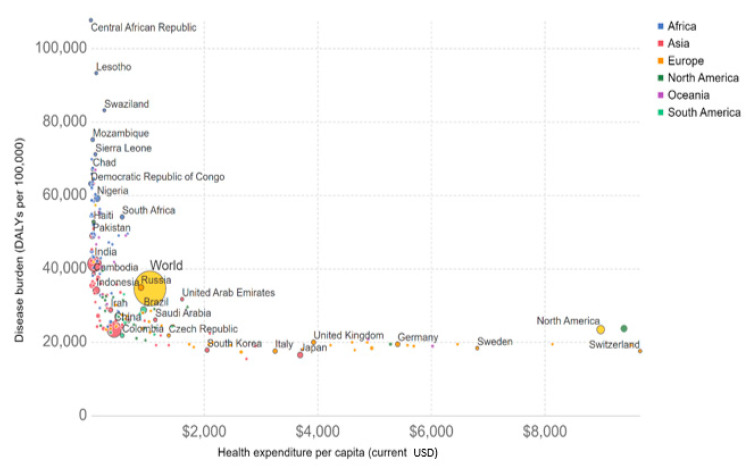
Relationship between the number of DALYs per 100,000 individuals vs. the health expenditure per capita in USD throughout the world in 2012. Information provided by M. Roser and H. Ritchie (2019) in “Causes of Death,” published online at OurWorldInData [87].

**Figure 22 ijerph-18-02532-f022:**
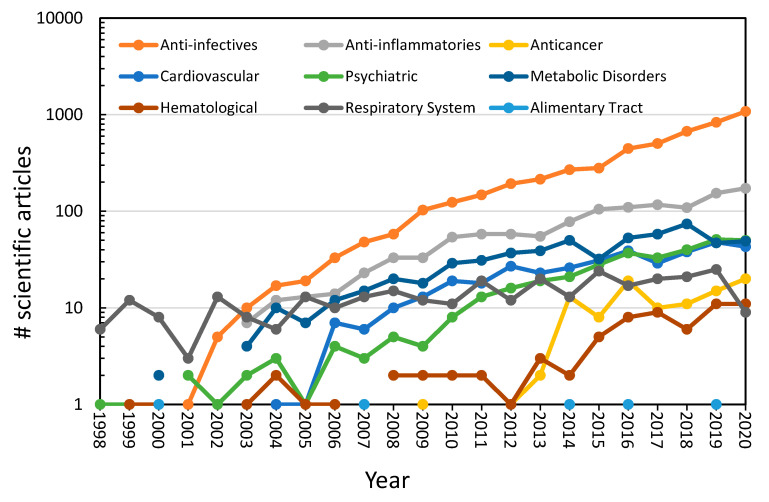
Research trends of pharmaceuticals in water and wastewater worldwide in the last 22 years. Data obtained from different research chains in Scopus as Table 1 describes.

**Figure 23 ijerph-18-02532-f023:**
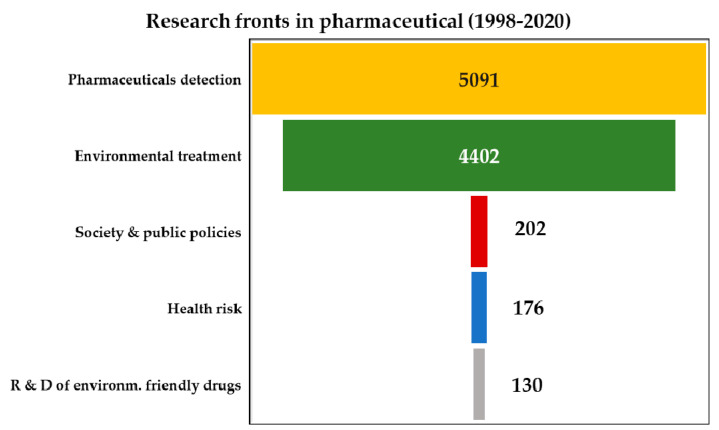
Number of scientific papers by pharmaceutical research fronts under study in Scopus according with the methodology used in Table 3. Data collected from 1998 to 2020.

**Figure 24 ijerph-18-02532-f024:**
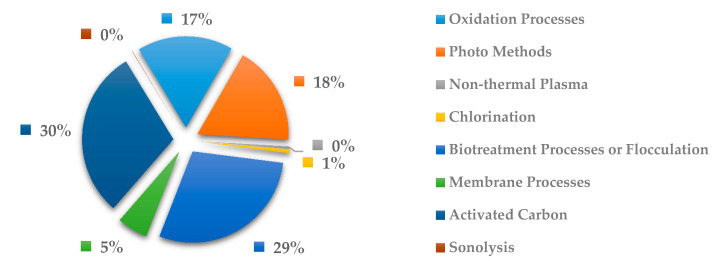
Percentage of scientific articles worldwide regarding wastewater treatments of anti-infective drugs between 1998 and 2020. Data collected in Scopus database by using wastewater or water as keywords and English as a language limitation.

**Figure 25 ijerph-18-02532-f025:**
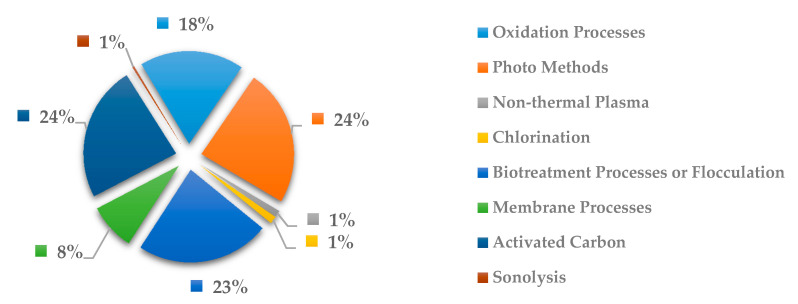
Percentage of scientific articles worldwide regarding wastewater treatments of anti-inflammatory drugs between 1998 and 2020. Data collected in Scopus database by using wastewater or water as keywords and English as a language limitation.

**Figure 26 ijerph-18-02532-f026:**
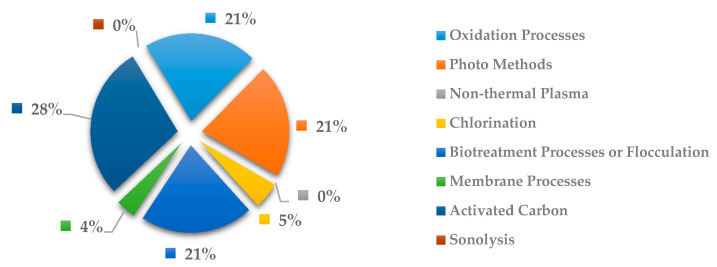
Percentage of scientific articles worldwide regarding wastewater treatments of anticancer drugs between 1998 and 2020. Data collected in Scopus database by using wastewater or water as keywords and English as a language limitation.

**Figure 27 ijerph-18-02532-f027:**
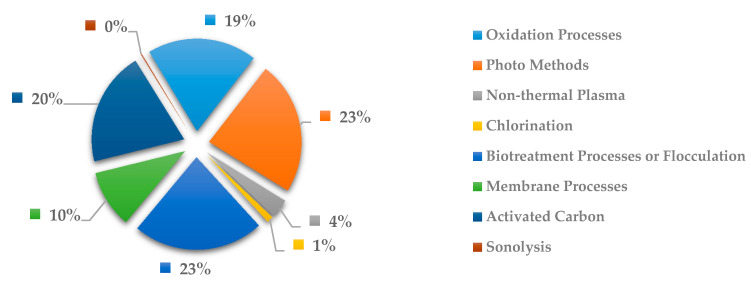
Percentage of scientific articles worldwide regarding wastewater treatments of cardiovascular drugs between 1998 and 2020. Data collected in Scopus database by using wastewater or water as keywords and English as a language limitation.

**Figure 28 ijerph-18-02532-f028:**
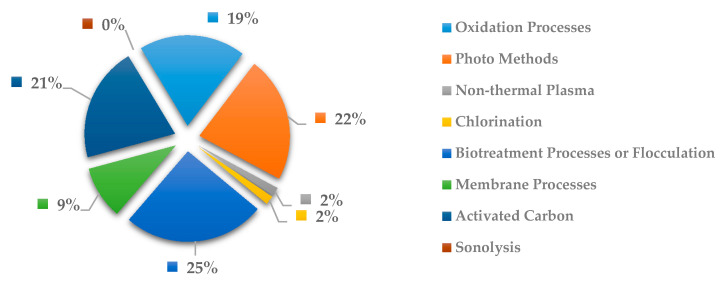
Percentage of scientific articles worldwide regarding wastewater treatments of psychiatric-neurological drugs between 1998 and 2020. Data collected in Scopus database by using wastewater or water as keywords and English as a language limitation.

**Figure 29 ijerph-18-02532-f029:**
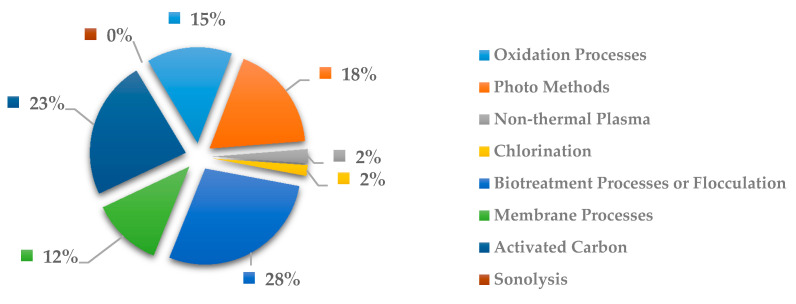
Percentage of scientific articles worldwide regarding wastewater treatments of metabolic disorder drugs between 1998 and 2020. Data collected in Scopus database by using wastewater or water as keywords and English as a language limitation.

**Figure 30 ijerph-18-02532-f030:**
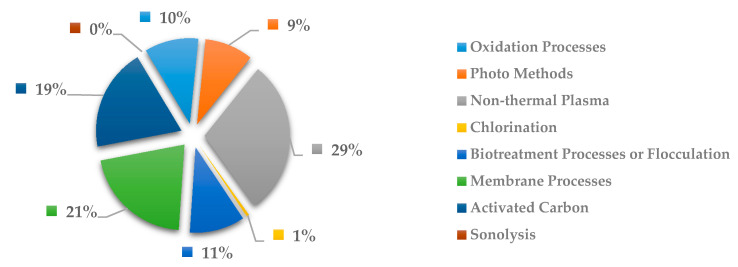
Percentage of scientific articles worldwide regarding wastewater treatments of hematological drugs between 1998 and 2020. Data collected in Scopus database by using wastewater or water as keywords and English as a language limitation.

**Figure 31 ijerph-18-02532-f031:**
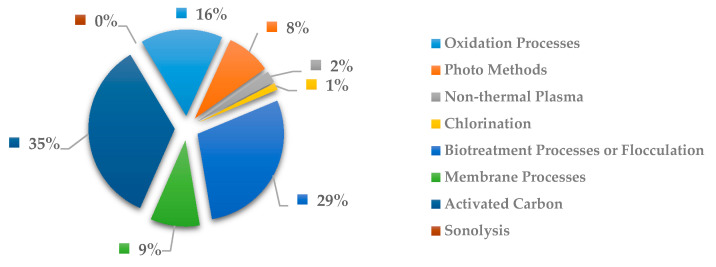
Percentage of scientific articles worldwide regarding wastewater treatments of respiratory system drugs between 1998 and 2020. Data collected in Scopus database by using wastewater or water as keywords and English as a language limitation.

**Figure 32 ijerph-18-02532-f032:**
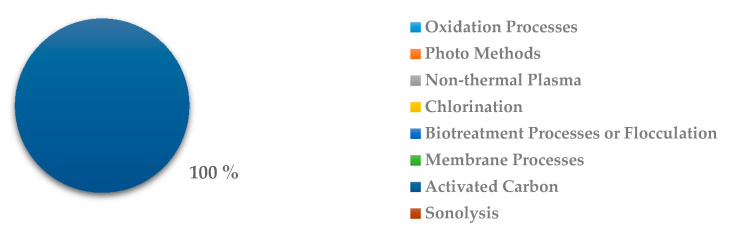
Percentage of scientific articles worldwide regarding wastewater treatments of alimentary tract and metabolism drugs between 1998 and 2020. Data collected in Scopus database by using wastewater or water as keywords and English as a language limitation.

**Table 1 ijerph-18-02532-t001:** Groups of diseases or health burden according to the Institute of Health Metrics and Evaluation (IHME). Information is from Max Roser and Hannah Ritchie [86], “Burden of Disease,” published online at OurWorldInData.org.

Communicable, Maternal, Neonatal, and Nutritional Diseases	Noncommunicable Diseases (NCDs)	Injuries
Diarrhea, lower respiratory and other common infectious diseases	Cardiovascular diseases (including stroke, heart disease and heart failure)	Road injuries
Neonatal disorders	Cancers	Other transport injuries
Maternal disorders	Respiratory disease	Falls
Malaria and neglected tropical diseases	Diabetes and blood and endocrine diseases	Drowning
Nutritional deficiencies	Mental and substance use disorders	Fire, heat, and hot substances
HIV/AIDS	Liver diseases	Poisonings
Tuberculosis	Digestive diseases	Self-harm
Other communicable diseases	Musculoskeletal disorders	Interpersonal violence
	Neurological disorders (including dementia)	Conflict and terrorism
	Other NCDs	Natural disasters

**Table 2 ijerph-18-02532-t002:** Keywords used in the pharmaceutical groups under study.

Anti-Infectives	Anti-Inflammatories	Anticancer	Cardiovascular System	Psychiatric—Neurological	Metabolic Disorders	Haematological Disease	Respiratory System	Alimentary Tract and Metabolism
Antibiotics	Cyclooxygenase (COX1)	Oncology	Antihypertensive drugs	Antidepressants	Antidiabetic	Blood-related drugs		
Antivirals	Cyclooxygenase (COX2)		Blood and blood-forming organs	Nervous system drugs	Thyroid drugs			
Fungus (pharmaceutical)	Musculoskeletal		Beta blocker		Pituitary drug			
	Analgesics		Agents acting on the renin-angiotensin system		Cholesterol-lowering			
	Opioids				Lipid-modifying agents			

**Table 3 ijerph-18-02532-t003:** Percentage distribution by geographical world regions of the research effort in terms of research articles for the removal of different pharmaceutical groups between 1998 to 2020 in Scopus using wastewater or water as keywords and English as a language limitation.

Pharmaceutical Group	Percentage				
	North America	Europe	Asia	Oceania	Africa
Anti-infectives	22%	11%	67%	0%	0%
Anti-inflammatories	21%	52%	27%	0%	0%
Anti-cancer	15%	33%	52%	0%	0%
Cardiovascular System	32%	46%	22%	0%	0%
Psychiatric—Neurological	39%	31%	30%	0%	0%
Metabolic disorder	36%	30%	27%	7%	0%
Haematological	39%	30%	26%	5%	0%
Respiratory System	45%	28%	27%	0%	0%
Alimentary Tract and metabolism	0%	50%	33%	17%	0%

**Table 4 ijerph-18-02532-t004:** Number of scientific articles wrote by the leading countries throughout the world regarding wastewater treatment of each pharmaceutical group between 1998 to 2020 in Scopus using wastewater or water as keywords and English as a language limitation.

Country	Number of Scientific Articles									
	Anti-Infectives	Anti-Inflammatories	Anti-Cancer	Cardiovascular System	Psychiatric—Neurological	Metabolic Disorder	Hematological	Respiratory System	Alimentary Tract and Metabolism	Total Scientific Articles per Country
China	1760	114	17	47	49	89	2	32	0	**2110**
United States	518	93	14	52	72	100	17	86	0	**952**
Spain	284	129	11	49	35	10	8	2	0	**528**
India	219	44	14	14	17	11	7	21	1	347
Germany	143	49	6	20	22	28	2	24	1	294
Iran	233	21	6	6	6	10	3	0	0	285
Brazil	164	54	6	16	14	14	1	0	0	269
United Kingdom	143	20	6	21	11	45	1	21	1	268
Italy	135	53	7	23	11	20	4	13	0	266
Canada	118	29	1	19	15	33	4	19	0	238
Australia	116	36	2	12	12	27	3	1	0	209
Japan	116	27	5	3	10	12	5	17	0	195
South Korea	131	32	2	7	4	17	1	0	0	194
Poland	90	28	6	20	15	6	0	0	0	165
Turkey	108	37	2	2	2	3	0	0	0	154
France	32	42	5	12	14	17	1	13	0	136
Mexico	51	30	1	8	1	9	1	11	0	112
Greece	52	18	3	10	6	4	3	1	1	97
Belgium	42	16	2	2	4	7	3	0	0	76
Finland	43	14	0	5	1	9	0	0	0	72
Israel	31	6	5	1	1	4	1	0	0	49
Total articles per pharmaceutical group	4529	892	121	349	322	475	67	261		
									4	

**Table 5 ijerph-18-02532-t005:** Pharmaceutical research fronts in under study in Scopus.

Main Research Fronts in Pharmaceuticals	Search Chain (Keywords)
1. Society and public policies (legislation for removal)	(Public policies) or (legislation) + pharmaceuticals + (water) or (wastewater) or (disposal)
2. Environmental treatment (removal)	(Removal) or (remotion) + (water) or (wastewater) + (pharmaceutical) or (emerging pollutants) + pharmaceuticals
3. Health risk	Health risk+ drinking water + pharmaceuticals
4. Detection of pharmaceuticals	Detection + (water) or (wastewater) + (pharmaceuticals) or (emerging pollutants) + pharmaceuticals
5. Development of environmentally friendly pharmaceuticals.	Environmentally friendly + pharmaceuticals + (water) or (wastewater)

**Table 6 ijerph-18-02532-t006:** Wastewater most well know regulations and their pharmaceutical approaches.

Legislation Source	Region Where It Regulates	Release Year	Remarks on Water Treatment Legislation in Pharmaceuticals Removal	Reference
Clean Water Act.	United States	1972	CWA oversees maintaining the physical, chemical, and biological integrity of national waters through the regulation and/or elimination of pollutants. However, the removal of bioaccumulated drugs that may represent a long-term problem is not specified.	[88]
Safe Drinking Water Act.	United States	1974	SDWA establishes national standards for drinking water and thus avoid the presence of human-made and naturally occurring contaminants. However, the removal of bioaccumulated drugs that may represent a long-term problem is not specified.	[89]
NOM-001-SEMARNAT-1996	México	1996	No observations regarding pharmaceuticals. Only organic compounds and inorganic chemicals.	[90]
Resource Conservation and Recovery Act.	United States	1976	RCRA regulates transportation, treatment, storage, and disposal of hazardous and non-hazardous solid waste. Among the non-hazardous are medical industrial waste. Nevertheless, it is not specified if those wastes are coming from pharmaceuticals.	[91]
Convention on the Protection and Use of Transboundary Watercourses and International Lakes.	International	1992	Management and protection of transboundary surface water. No observations regarding pharmaceuticals.	[92]
The Basel Convention on the Control of Transboundary Movements of Hazardous Wastes and their Disposal.	International (OECD countries)	1989	Wastes to be controlled inside this legislation are waste from pharmaceuticals, medical care hospitals, chemicals used in formulations, inorganic compounds, organic compounds, etc. Additionally, it is important to note that “pharmaceutical wastes” are treated as a whole, instead of being dissected in each of the different pharmaceutical groups.	[93]
Bamako Convention on the Ban of the Import into Africa and the Control of Transboundary Movement and Management of Hazardous Wastes within Africa.	International	1991	Clinical waste from medical facilities, organic and inorganic compounds, pharmaceuticals, anything related to the production of drugs, etc. Additionally, it is important to note that “pharmaceutical wastes” are treated as a whole, instead of being dissected in each of the different pharmaceutical groups.	[94]
Rotterdam Convention on the Prior Informed Consent Procedure for Certain Hazardous Chemicals and Pesticides in International Trade.	International	1998	Regulations regarding any type of pesticides and its production. No observations on pharmaceutical wastes.	[95]
Convention on the Transboundary Effects of Industrial Accidents.	International	1992	No specific remarks on pharmaceutical waste. Legislation related to the containment of hazardous substances (organic and inorganic chemicals) released after industrial accidents and its methods to reduce their potential risk on human health and the environment.	[96]
European Agreement concerning the International Carriage of Dangerous Goods by Inland Waterways.	European Union	2000	No specific remarks on pharmaceutical waste or wastes of any kind. Only ensures the carrying of dangerous goods such as: flammable liquids, flammable solids, radioactive material, toxic and infectious substances, etc. Additionally, contributes to the protection of the environment by preventing any pollution from the carriage of goods.	[97]
International Code of Conduct on the Distribution and Use of Pesticides.	International	1985	Registration of pesticides with the responsible national government Additionally, emphasizes that even existing pesticides should be revised to meet the safety requirements. No specific remarks on pharmaceutical waste or wastes of any kind.	[98]
Minamata Convention on Mercury.	International	2013	No observations on any type of pharmaceutical waste. The Minamata Convention seeks to reduce (or eliminate) the use of mercury from artisanal and small-scale gold mining. Additionally, the reduction of mercury in manufacturing processes of any kind.	[99]
Stockholm Convention on Persistent Organic Pollutants.	International	2001	No observations on any type of pharmaceutical waste. The Stockholm Convention seeks to protect the human health and the environment from persistent organic pollutants.	[100]
Waigani Convention.	South Pacific Region	1995	No observations on any type of pharmaceutical waste. The Waigani Convention is similar to the Basel convention but with a focus on radioactive waste and its control, movement, and production.	[101]

**Table 7 ijerph-18-02532-t007:** Pharmaceutical removal methods in pharmaceutical groups under study in Scopus.

Methods of Removal	Keywords
Oxidation processes	(Oxidation) or (advance oxidation) or (Fenton).
Photo methods	(UV) or (photo) or (ozonation) or (solar).
Nonthermal plasma	(Plasma).
Chlorination	(Chlorination).
Biotreatments	(Biotreatment) or (flocculation) or (biodegradation) or (nitrification) or (bioreactor).
Membranes	(Membrane) or (osmosis).
Activated carbon	(Activated carbon) or (carbon)
Sonolysis	Sonolysis.

## Data Availability

Data is contained within the article.

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
