# Peer review of "Pharmaceuticals Market, Consumption Trends and Disease Incidence Are Not Driving the Pharmaceutical Research on Water and Wastewater"

_ijerph, 2021, doi:10.3390/ijerph18052532_

Round 1

Reviewer 1 Report

The paper is informative and generally well written. However, the theme of the paper is not very clear. Please find my detailed comments below:

  1. Please double check the accuracy of revenue of pharmaceutical indutry (Line 37-40) and cite reference
  2. Typo at Line 54 and 55
  3. In Introduction section, the market and consumption part can be trimmed or removed to section 2. In addition, authors should elaborate more about the environmental hazard pharmaceutical industry produced, such as what type of hazard it usually produce (chemical, bio, etc.), geographic distribution of these pollution, types of pollution (water is introduced, how about soil and air?), etc.
  4. Overall, the paper seems not logically flow well. The two big topics, market/trend & research front and regulation & disposal seems unrelated. The author should consider exclude one of the topics and only focus on one, otherwise the author will need to find out a logic linkage between them.
  5. Please update information in Figures to the year of 2020, at least 2019
  6. It would be helpful to elucidate the source of data/information for your figures.
  7. Please double check the correctness of information in the figures. For example, Eylea is developed by Regeneron, not Bayer/Regeneron. There are several such mistakes.
  8. There are several inconsistent information throughout the paper. Please double check
  9. Why OECD countries are specifically investigated?
  10. Figure 9 is overlaid.
  11. How 'removal of pharmaceutical groups in water and wastewater' can represent the research of pharmaceuticals? It is a very fundamental concept in your paper and must be justified.
  12. Suggest to improve the aesthetic appearance of Figures 23-30. And suggest to combine these figures into one figure or table. Same thing with figures in sections 4.2 and 6.

Author Response

Dear Editor and Reviewers, 
We are grateful for the prompt and careful reading of our manuscript as well as the insightful comments. 
In general, we improved the use of English language. In the following, we address in detail each comment from the reviewers; the changes can be seen within the document in word in the menu in the section of (review) and with the tool of (track changes) in the new version of the manuscript for the sake of readability. However, additional changes can be observed in the blue text. 

We hope that the new version of our manuscript addresses the concerns of the reviewers and secures publication in Sustainability.

I remain grateful for your support on behalf of the authors.

Below are our responses to comments made by reviewer 1:

The paper is informative and generally well written. However, the theme of the paper is not very clear. Please find my detailed comments below:

  1. Please double check the accuracy of revenue of pharmaceutical indutry (Line 37-40) and cite reference.

[Answer]: The amounts were checked and corrected.

2. Typo at Line 54 and 55

[Answer]: it was the number 2 after the percentage in 11.6%, therefore it was removed.

3. In Introduction section, the market and consumption part can be trimmed or removed to section 2. In addition, authors should elaborate more about the environmental hazard pharmaceutical industry produced, such as what type of hazard it usually produce (chemical, bio, etc.), geographic distribution of these pollution, types of pollution (water is introduced, how about soil and air?), etc.

[Answer]: We understand the importance of the environmental hazards generated by the  pharmaceutical industry and have to be exposed and discussed into the scientific community; however, the scope of this review is to analyze the relationships among the pharmaceuticals market, the consumption trends, the fatal disease incidence rate and disease burden in the population with respect to the different pharmaceuticals research fronts with the aim of finding the drivers and interests that motivate and promote the research on certain type of pharmaceuticals. Furthermore, the findings are revealing given that the information presented has shown dissident and coinciding actions to achieve sustainability in wastewater treatment by removing drugs.

4. Overall, the paper seems not logically flow well. The two big topics, market/trend & research front and regulation & disposal seems unrelated. The author should consider exclude one of the topics and only focus on one, otherwise the author will need to find out a logic linkage between them.

[Answer]: As mentioned previously, the novelty and aim of the review is an attempt to find the drivers and interests that motivate and promote the research on certain type of pharmaceuticals. Thus,  the pharmaceuticals market, the consumption trends, the fatal disease incidence rate and disease burden in the population were analyzed as potential drivers or interests that motivate and determine the research work on certain groups of pharmaceuticals in wastewater. Thus, the manuscript was revised and improved with the aim of emphasizing the relationship or mismatch among market/trends and research fronts and regulation & disposal.

5. Please update information in Figures to the year of 2020, at least 2019

[Answer]: Most of the figures in sections 2, 3, 4, 5, and 6 were updated based on the 2020 reports. 

6. It would be helpful to elucidate the source of data/information for your figures.

[Answer]: The figures have been identified by its corresponding source as well as in the manuscript it is indicated the sources uses to the information presented in figures.

7. Please double check the correctness of information in the figures. For example, Eylea is developed by Regeneron, not Bayer/Regeneron. There are several such mistakes.

[Answer]: The figures have been checked.

8. There are several inconsistent information throughout the paper.

[Answer]: The manuscript has been reviewed and improved.

9. Why OECD countries are specifically investigated?

[Answer]: OECD countries are not specifically investigated, we tried to gather information and know the pharmaceutical consumption status worldwide; however, information of non OECD countries is difficult to obtain or is not available. Therefore, the analysis and discussion of the data presented in the manuscript is considered, in some how, to be representative the pharmaceutical consumption world.  Furthermore, market, consumption patterns and development of wastewater treatment technologies in OECD countries normally drives the international research agenda.

10. Figure 9 is overlaid.

[Answer]: The figure 9 has been fixed.

11. How 'removal of pharmaceutical groups in water and wastewater' can represent the research of pharmaceuticals? It is a very fundamental concept in your paper and must be justified.

[Answer]: It is clear that research on pharmaceuticals in water and wastewater can not represent the total pharmaceuticals research picture, but the aim of this review is to elucidate if the market, consumption patterns of drugs and diseases faced by human race drives and motivates the pharmaceutical research on water and wastewater.  It was a challenging work to categorize and integrate the different pharmaceuticals groups because the market, the medical sector and the environmental research community have different criteria to classified them. Even though such difficulties, we consider the research of pharmaceutical in water and wastewater can represent in certain level the global research status of pharmaceuticals.

12. Suggest to improve the aesthetic appearance of Figures 23-30. And suggest to combine these figures into one figure or table. Same thing with figures in sections 4.2 and 6.

[Answer]: Figures 23 to 30 have been combined in table 3. Likewise, the figures 31 to 38 have been combined in Table 4. However, in section 6, we consider more convenient to show an analysis of the data by presenting pie charts due the nature of the information in which it is simple to find divergences or similitudes on the different methodologies and technologies for drug removal in wastewater.

Reviewer 2 Report

Congratulations to the authors of the development of a reliable and comprehensive compendium of the latest medical knowledge in the field of the analysis of pharmaceutical market, drugs consumption trends and the pharmaceutical research interests worldwide. This paper also presents percentage distributions of the research efforts in producing a scientific study according to different wastewater treatments for each pharmaceutical group. 

Below are some suggestions to improve the quality of the presented work.

Minor revisions:

  1. Please correct figure 9, it looks like two separate figure are overlapping at the same time.
  2. Tables No. 2 should be corrected. For better readability, I propose to present the tables in horizontal rather than vertical oreintment.
  3. I propose on issue 4.1 Research trends by continent.
    instead of 8 pie charts, present the data in one summary table. This will enable the reader to better analyze the studied problem with the scope of research efforts conducted on pharmaceuticals groups in water and
    wastewater in different regions of the globe.

Author Response

Dear Editor and Reviewers, 
We are grateful for the prompt and careful reading of our manuscript as well as the insightful comments. 
In general, we improved the use of English language. In the following, we address in detail each comment from the reviewers; the changes can be seen within the document in word in the menu in the section of (review) and with the tool of (track changes) in the new version of the manuscript for the sake of readability. However, additional changes can be observed in the blue text. 

We hope that the new version of our manuscript addresses the concerns of the reviewers and secures publication in Sustainability.

I remain grateful for your support on behalf of the authors.

Below are our responses to comments made by reviewer 2:

Congratulations to the authors of the development of a reliable and comprehensive compendium of the latest medical knowledge in the field of the analysis of pharmaceutical market, drugs consumption trends and the pharmaceutical research interests worldwide. This paper also presents percentage distributions of the research efforts in producing a scientific study according to different wastewater treatments for each pharmaceutical group. 

 Below are some suggestions to improve the quality of the presented work.

Minor revisions:

1.   Please correct figure 9, it looks like two separate figure are overlapping at the same time.

[Answer]: The figure 9 has been fixed.

2.   Tables No. 2 should be corrected. For better readability, I propose to present the tables in horizontal rather than vertical oreintment.

[Answer]: Table 2 has been fixed; thus, it is shown now in horizontal form.

3.   I propose on issue 4.1 Research trends by continent.

instead of 8 pie charts, present the data in one summary table. This will enable the reader to better analyze the studied problem with the scope of research efforts conducted on pharmaceuticals groups in water and wastewater in different regions of the globe.

[Answer]: Figures 23 to 30 have been combined in table 3. Likewise, the figures 31 to 38 have been combined in table 4.

Reviewer 3 Report

1. This manuscript includes figures and tables that appear to be obtained from other sources. It would be better to create your own figures and tables with proper attribution for sources.

2. The manuscript describes pharmaceuticals as organic compounds. Inorganic compounds are used as pharmaceuticals as well. 

3. The paper lacks focus. As such, it reads like an information source or report that could be found on-line. For making a contribution to the literature, the authors could consider a focus area and develop a review paper on that. As it is now, the document would be more suited as a report on a web site.

4. The paper devotes quite a bit of space to wastewater issues. That could be an interesting domain of focus.

Author Response

Dear Editor and Reviewers,

We are grateful for the prompt and careful reading of our manuscript as well as the insightful comments.

In general, we improved the use of English language. In the following, we address in detail each comment from the reviewers; the changes can be seen within the document in word in the menu in the section of (review) and with the tool of (track changes) in the new version of the manuscript for the sake of readability.

We hope that the new version of our manuscript addresses the concerns of the reviewers and secures publication in Sustainability.

I remain grateful for your support on behalf of the authors

Below are our responses to comments made by reviewer 3:

  1. This manuscript includes figures and tables that appear to be obtained from other sources. It would be better to create your own figures and tables with proper attribution for sources.

[Answer]: All figures have been created by us. But figure 20 and 21 cannot be created by us with precision. As a result, we have permission to use them due to the source correspond to the world census in which it is a resource available and free of restrictions.

  1. The manuscript describes pharmaceuticals as organic compounds. Inorganic compounds are used as pharmaceuticals as well. 

[Answer]:

Indeed, some drugs are also inorganic; however, most of the pharmaceuticals are organic; thus, this review only discusses organic compounds. Likewise, to have a potential traceability of the drugs that reach the wastewater, the information has to be associated with the commercial registration of the drugs to obtain the quantity sold and consumed. In addition, drug removal studies in wastewater are described in terms of the market name.

  1. The paper lacks focus. As such, it reads like an information source or report that could be found on-line. For making a contribution to the literature, the authors could consider a focus area and develop a review paper on that. As it is now, the document would be more suited as a report on a web site.

[Answer]: The manuscript was revised and improved to be clearer and focused. In fact, most of the information is found in different sources. However, the attempt to analyze the relationships among the pharmaceuticals market, the consumption trends, the fatal disease incidence rate and disease burden in the population with respect to the different pharmaceuticals research fronts is the main contribution of the review. The study reveals a growing global pharmaceutical market, a continuing increase of pharmaceutical consumption and an intensive research work on different pharmaceuticals fronts; however, such research is not totally aligned with the market trends and consumption patterns. Furthermore, the legislation and regulations available do not address the treatment and disposal of emergent contaminants, such as the pharmaceuticals; neither these legislations or regulations promote incentives in favor of the pharmaceutical industry to generate new drugs that can provide adequate treatment for the patient, but also that the drugs have to be friendly to the environment under a philosophy of green chemistry.

  1. The paper devotes quite a bit of space to wastewater issues. That could be an interesting domain of focus.

[Answer]: As mentioned previously, the aim of the review is an attempt to find the drivers and interests that motivate and promote the research on certain type of pharmaceuticals. Thus,  the pharmaceuticals market, the consumption trends, the fatal disease incidence rate and disease burden in the population were analyzed as the potential drivers or interests that motivate and determine the research work on certain groups of pharmaceuticals on water and wastewater. The manuscript was revised and improved with the aim of emphasizing the relationship or mismatch among market/trends and research fronts and regulation & disposal.

Round 2

Reviewer 1 Report

Thanks for the correction.

Author Response

Below we show the observations made by the reviewer:

[REVIEWER 1]: Thanks for the correction.

[Answer]: Thank you for the kind and careful revision on this review article. In the updated revision on the manuscript the title has been changed in the following way: Pharmaceuticals market, consumption trends and disease incidence are not driving the pharmaceutical research on Water and Wastewater.

Reviewer 3 Report

The authors address the suggestions made by this reviewer.

The title is still quite confusing and the purpose of the paper should be made clear to a reader and then a consistent thread of thought should be used through the whole paper.

Author Response

Below we show the observations made by the reviewer:

Comments and Suggestions for Authors:

The authors address the suggestions made by this reviewer.

[Reviewer 3]: The title is still quite confusing and the purpose of the paper should be made clear to a reader and then a consistent thread of thought should be used through the whole paper.

[Answer]: we appreciate the helpful and careful revision on this review article. In the updated revision on the manuscript the title has been changed in the following way: Pharmaceuticals market, consumption trends and disease incidence are not driving the pharmaceutical research on Water and Wastewater.